# Leptin mediates the regulation of muscle mass and strength by adipose tissue

Kelsey H. Collins[1,2,3], Chang Gui[4,5], Erica V. Ely[1,2,3,4], Kristin L. Lenz[1,2,3], Charles A. Harris[6], Farshid Guilak[1,2,3,4] and Gretchen A. Meyer[1,3,4,5,7] (ID)

[1]*Department of Orthopaedic Surgery, Washington University in St. Louis, MO, USA*
[2]*Shriners Hospitals for Children, St Louis, MO, USA*
[3]*Center of Regenerative Medicine, Washington University in St. Louis, MO, USA*
[4]*Department of Biomedical Engineering, Washington University in St. Louis, MO, USA*
[5]*Program in Physical Therapy, Washington University, St Louis, MO, USA*
[6]*Division of Endocrinology, Metabolism & Lipid Research, Washington University, St Louis, Missouri, USA*
[7]*Department of Neurology, Washington University in St. Louis, St Louis, MO, USA*

Handling Editors: Scott Powers & Troy Hornberger

The peer review history is available in the Supporting Information section of this article (https://doi.org/10.1113/JP283034#support-information-section).

**Abstract** Adipose tissue secretes numerous cytokines (termed 'adipokines') that have known or hypothesized actions on skeletal muscle. The majority of adipokines have been implicated in the pathological link between excess adipose and muscle insulin resistance, but approximately half also have documented *in vitro* effects on myogenesis and/or hypertrophy. This complexity suggests

**Kelsey Collins** is a research instructor in the laboratory of Dr Farshid Guilak in the Department of Orthopaedic Surgery at Washington University in St Louis. She received her PhD in Biomedical Engineering from the University of Calgary in 2017. She uses tissue engineering and preclinical models to dissect the role of adipose tissue signalling in musculoskeletal damage. During her postdoctoral training, she and Dr Guilak initiated collaboration with Dr Gretchen Meyer in Physical Therapy to understand the role of leptin in adipose–muscle crosstalk in a fat-free lipodystrophic mouse model. She plans to begin her independent research laboratory supported by her NIH NIAMS K99/R00 Pathway to Independence Award in the Spring of 2023.

The Journal of Physiology

a potential dual role for adipokines in the regulation of muscle mass in homeostasis and the development of pathology. In this study, we used lipodystrophic 'fat-free' mice to demonstrate that adipose tissue is indeed necessary for the development of normal muscle mass and strength. Fat-free mice had significantly reduced mass (~15%) and peak contractile tension (~20%) of fast-twitch muscles, a slowing of contractile dynamics and decreased cross-sectional area of fast twitch fibres compared to wild-type littermates. These deficits in mass and contractile tension were fully rescued by reconstitution of ~10% of normal adipose mass, indicating that this phenotype is the direct consequence of absent adipose. We then showed that the rescue is solely mediated by the adipokine leptin, as similar reconstitution of adipose from leptin-knockout mice fails to rescue mass or strength. Together, these data indicate that the development of muscle mass and strength in wild-type mice is dependent on adipose-secreted leptin. This finding extends our current understanding of the multiple roles of adipokines in physiology as well as disease pathophysiology to include a critical role for the adipokine leptin in muscle homeostasis.

(Received 25 February 2022; accepted after revision 4 July 2022; first published online 17 July 2022)

**Corresponding author** G. A. Meyer: Program in Physical Therapy, Washington University in St. Louis, 4444 Forest Park Ave, St. Louis, MO 63108, USA.    Email: meyerg@wustl.edu

**Abstract figure legend** Adipokines help maintain healthy muscle physiology in wild-type (WT) mice (far left panel). Mice lacking adipose tissue (fat-free (FF) mice) experience muscle atrophy and contractile dysfunction (middle left panel). Add-back of a small quantity of adipose tissue to FF mice can fully restore WT muscle physiology (middle right panel). However, add-back of adipose tissue lacking the adipokine leptin fails to rescue the FF muscle phenotype (far right panel). Therefore we conclude that adipose-derived leptin is required to develop and maintain full muscle mass and strength.

## Key points

- Adipose-derived cytokines (adipokines) have long been implicated in the pathogenesis of insulin resistance in obesity but likely have other under-appreciated roles in muscle physiology.
- Here we use a fat-free mouse to show that adipose tissue is necessary for the normal development of muscle mass and strength.
- Through add-back of genetically modified adipose tissue we show that leptin is the key adipokine mediating this regulation.
- This expands our understanding of leptin's role in adipose–muscle signalling to include development and homeostasis and adds the surprising finding that leptin is the sole mediator of the maintenance of muscle mass and strength by adipose tissue.

## Introduction

Adipose tissue and skeletal muscle share a metabolic link that fuels muscle contraction and regulates energy storage. Research over the past several decades has expanded this link into a multi-faceted model that includes metabolic substrates, adipose-derived cyto-kines (a.k.a. adipokines) and inflammatory mediators (reviewed in Sell, Dietze-Schroeder, & Eckel, 2006). Currently, there are more than 20 adipokines with known or hypothesized action on skeletal muscle (Li et al., 2017; Nicholson et al., 2018). These have pre-dominantly been studied in the context of obesity and in the pathogenesis of diabetes, where the altered secretion of adipokines contributes to insulin resistance and metabolic dysfunction (Nicholson et al., 2018).

However, many adipokines are pleiotropic, affecting multiple pathways and cell types in context-dependent ways. For example, adipokines such as adiponectin, visfatin and pre-adipocyte factor 1 (pref-1) promote *in vitro* myogenesis (Krzysik-Walker et al., 2011; Shin et al., 2014; Yu et al., 2021) and leptin, chemerin and fibroblast growth factor 21 (FGF-21) activate the Akt/mechanistic target of rapamycin (mTOR) signalling cascade (a major muscle anabolic pathway) (Minard et al., 2016; Rodríguez et al., 2011; Yang et al., 2012). This evidence points to a role for adipokines in regulating muscle development and homeostasis. This hypothesis is supported by considerable muscle pathology documented in individuals with deficiency of adipose tissue due to lipodystrophy (Akinci et al., 2017; Garg et al., 2000), but

has yet to be mechanistically dissected to uncover the causative player(s).

A number of studies have shown strong, albeit indirect, evidence to support a role for leptin in regulating muscle mass and strength (Rosenbaum et al., 2005). For example, mice with leptin loss of function (*ob/ob*) have lower mass of fast-fibred muscles with reduced contractile forces (Bruton et al., 2002). The reduction in mass can be partially rescued by 4 weeks of leptin treatment at physiological levels (Sainz et al., 2009), and this finding has been replicated *in vitro* where a direct action of leptin on muscle anabolic and catabolic pathways has been described (Kellerer et al., 1997; Sainz et al., 2009). However, other adipokines have also been implicated with similar findings. Knockout of adiponectin abrogates the exercise-induced increase in muscle mass and strength in mice, suggesting it plays a role in hypertrophy (Inoue et al., 2017). Knockout of visfatin in muscle significantly reduces muscle mass and strength (Frederick et al., 2016), suggesting it plays an important role in muscle development and function. These findings support a likely coordinated action of multiple adipokines in maintaining muscle mass and strength.

The majority of adipokines are not exclusively produced by adipose tissue, adding another layer of factors to disentangle to directly triangulate the role of adipokines in muscle pathology. Many pro-inflammatory adipokines are also secreted by macrophages (Guilherme et al., 2019), and leptin, adiponectin and visfatin are all also produced by the muscle itself, where these factors are thought to exert autocrine action (Frederick et al., 2016; Krause et al., 2008; Wolsk et al., 2012). These findings have complicated interpretation of global knockout models such as the *ob/ob* mouse since the action of adipose-secreted leptin, for example, and cannot be precisely separated from leptin derived from sources other than adipose tissue (e.g. skeletal muscle; Wang et al., 1998). Furthermore, the actions of most adipokines are pleiotropic, such that knockout affects the homeostasis of multiple tissues. In the *ob/ob* mouse, for example, knockout of leptin affects its action on adipose tissue, the brain and the adrenal glands causing morbid obesity, dyslipidaemia, and elevated glucocorticoid signalling (Livingstone et al., 2009) that are likely to compound the direct action of leptin on muscle by potentially affecting muscle through indirect signalling pathways.

In this study we sought to dissect the role of adipokines, and specifically adipose-secreted leptin, in regulating muscle physiology. We utilized a constitutive complete lipodystrophic, or 'fat-free' mouse model to demonstrate that adipose tissue is required for the maintenance of muscle mass and strength. We then used tissue engineering fat transplantation approaches to show that only a fraction of typical whole body adipose is required for muscle maintenance, and the effect is mediated by adipose-secreted leptin. We demonstrate that restoration of muscle mass and strength by adipose is separable from insulin resistance and other factors known to play a role in muscle pathology, indicating that a small amount of circulating leptin can maintain muscle mass and function independent of metabolic disturbance.

## Methods

### Ethical approval

All procedures were performed in accordance with the National Institutes of Health's *Guide for the Use and Care of Laboratory Animals* and were approved by the Animal Studies Committee of the Washington University School of Medicine (IACUC 20-0459 and 19-0774). The investigators conducted these procedures in accordance with the ARRIVE (Animal Research: Reporting of *In Vivo* Experiments) Guidelines 2.0.

### Animals

Experiments were performed on male fat-free (FF) and wild-type (WT) littermate control mice on a C57BL/6J background. FF mice were generated by crossing adiponectin-Cre mice (The Jackson Laboratory, Bar Harbor, ME, USA: 028020) with lox-stop-lox-Rosa diphtheria toxin mice (The Jackson Laboratory: 010527) (Collins et al., 2020; Wu et al., 2018). The resulting FF mice constitutively have a complete absence of adipose tissue from birth. These mice and DTA/+ littermate WT control were maintained at thermoneutrality (30°C) on a chow diet (10% fat) with free cage activity. Exact animal numbers used in each experiments are listed in the figure legends. Complement factor D constitutive knockout mice were used to determine the role of adipsin in the FF muscle physiology phenotype (a generous gift from Drs Atkinson and Wu; Wu et al., 2018).

### Mouse embryonic fibroblast transplantation

A subset of FF mice were treated with mouse embryonic fibroblasts (MEFs) at between 3 and 5 weeks of age. MEFs were isolated either from WT, leptin heterozygous (*ob/+*) or leptin homozygous knockout (*ob/ob*) pups as previously described (Ferguson et al., 2018). To obtain *ob/+* and *ob/ob* pups at a 50/50 Mendelian ratio, we first treated constitutive leptin (*ob*) homozygous knockout mice (*ob/ob*, Jax ID: 00632) with WT MEFs as leptin is required for pregnancy (Brenot et al., 2020). Once treated, *ob/ob* MEF sires were mated with a heterozygous *ob/+* dam. MEFs were genotyped at implantation. Briefly, for all MEF preparations, 14 days post-identification of a copulation plug, or presumptive embryonic day 14, the pregnant dam was euthanized, and pups were

isolated, genotyped and prepared for delivery to FF mice as MEF cell-based injections. Genotypes were identified as previously described (Ferguson et al., 2018). Remaining tissues were minced and digested in 1.25 ml of 0.05% trypsin for 45 min to 1 h at 37°C. Trypsin was neutralized with a mixture of Dulbecco's modified Eagle's medium, 1% penicillin–streptomycin and 10% fetal bovine serum. The digested tissue was dissociated to a single cell suspension by vigorous pipetting and filtered through a 70 $\mu$m cell strainer. The filtrate was then centrifuged at $500 \times g$ for 6 min, and the cell pellet was resuspended in 250 $\mu$l of sterile phosphate-buffered saline (PBS). The injection was delivered subcutaneously to the sternal aspect of a donor FF mouse (3–5 weeks old) under 2% isoflurane with a 27 gauge needle. Injected MEFs generated a consistent small subcutaneous adipose tissue depot ($\sim$1 g of tissue, 10–15% of WT total adipose tissue mass).

### Adrenalectomy

A subset of FF mice were treated with surgical adrenalectomy, or removal of both adrenal glands, at 5 weeks of age to evaluate the relationship between glucocorticoids and muscle mass. Mice were continuously anaesthetized with 2% inhaled isoflurane. An $\sim$5 mm incision was made through the skin on the back just inferior to the rib cage. A second incision was made through the abdominal wall to access the visceral cavity. The adrenal gland was accessed through gentle retraction of surrounding tissues and excised with a pair of ringed forceps. The muscle incision was closed with 5-0 braided silk suture and the skin incision was closed with Vetbond suture glue. The procedure was repeated on the other side for bilateral adrenalectomy and then mice were allowed to recover with analgesia (5–10 mg/kg meloxicam) individually in a pre-warmed cage.

### Exogenous leptin delivery

Another subset of FF mice were given exogenous mouse recombinant leptin (1 mg/kg body weight; R&D Systems, Minneapolis, MN, USA) or saline by I.P. injection daily for 7 days. Then, 30 min prior to sacrifice, mice were given puromycin (0.04 $\mu$mol/g body weight) by I.P. injection. This technique enables protein synthesis rates to be estimated from puromycin quantification in muscle western blots with good correlation to standard radiolabelling methods (Goodman & Hornberger, 2013). Untreated littermates (WT) were used as controls for this experiment.

### Insulin- and glucose-tolerance tests

Insulin- and glucose-tolerance tests were performed at 14 and 15 weeks of age after fasting mice for 4–6 h (Collins et al., 2020). Baseline fasting glucose levels were measured by tail bleed at 0 h. To determine insulin tolerance, 0.75 U/kg body mass of insulin (Humulin R diluted to 75 mU/ml in sterile PBS, 1% volume/body mass) was administered by I.P. injection. For glucose-tolerance tests, animals were challenged with 1 g/kg dextrose (10% dextrose in sterile water: 1% volume/body mass) by I.P. injection. Serial blood glucose measurements were taken via tail vein at 20, 40, 60 and 120 min after injection using commercially available glucose strips read by a glucometer (Contour; Bayer, Leverkusen, Germany). Area under the curve (AUC) was calculated for each test.

All mice were sacrificed at 6 or 16 weeks for acute muscle physiology testing and morphological measurements. Serum was collected immediately after sacrifice via retroorbital puncture following which tissues were prepared for further analysis as described for each assay.

### Muscle contraction assessment

Under anaesthesia (2% inhaled isoflurane at 2 l/min) fifth toe extensor digitorum longus (EDL) and soleus (SOL) muscles were isolated for *ex vivo* contractile assessment (Biltz et al., 2020). Anaesthetic plane was assessed every 5 min by toe-pinch, and following dissections mice were euthanized by cervical dislocation. The distal end of each muscle was secured to a dual mode force–length ergometer and the proximal end to a fixed post within a muscle stimulation system (1200A Aurora Scientific, Aurora, Ontario, Canada). Each muscle was completely submerged in mammalian Ringer's solution (mM: 137 NaCl, 5 KCl, 2 $CaCl_2$, 1 $MgSO_4$, 1 $NaH_2PO_4$, 24 $NaHCO_3$, 11 glucose containing 10 mg/l curare) and maintained at 37°C. The experimenter running the test was blinded to the treatment and genotype of each animal. Muscle activation was achieved through an electrical stimulator (701C, Aurora Scientific) with parallel platinum plate electrodes that extended the length of the muscle. Optimal muscle length was determined by increasing muscle length by 10% of slack fibre length until twitch forces plateaued and then increasing by 5% slack fibre length until tetanic forces plateaued. At optimal muscle length, an isometric tetanic (300 ms train of 0.3 ms pulses at 225 Hz) and twitch contraction were recorded. Next, fatigue was evaluated with repeated tetanic contractions every 20 s (EDL) or 10 s (SOL) until force fell below 50% of the peak isometric tetanus. Fibre length was measured using a microscope reticule at optimal length and muscle mass was measured post-testing. These measures were used with published values for pennation angle and muscle density to compute physiological cross-sectional area (PCSA) (Burkholder et al., 1994). Peak forces were normalized to PCSA to account for differences in muscle size.

## *Ex vivo* leptin incubation

EDL muscles were also isolated from saline and leptin treated mice to compare the effects of acute *ex vivo* and chronic *in vivo* leptin exposure. EDLs from saline treated mice were incubated in Ringer's solution supplemented with 0.01 mg/ml mouse recombinant leptin while the contralateral EDL was simultaneously incubated in Ringer's solution only. Twitch contractions were elicited and recorded at initial estimation of optimal muscle length (this time determined from twitch contractions alone) and at 5, 15 and 30 min of incubation. No tetanic contractions were elicited to avoid activating protein synthetic pathways. These muscles were then removed, weighed and flash-frozen for western blotting. EDLs from leptin treated mice were also assessed to determine the chronic effects of *in vivo* leptin exposure on twitch contractions.

### Histological assessment

Gastrocnemius, soleus and plantaris muscles were flash-frozen together in liquid nitrogen-cooled isopentane for histological assessment of fibre areas and fibre types. Axial sections of 10 $\mu$m were cut from the midbelly and immunostained against type 1, type 2a and type 2b myosin heavy chain isoforms (Developmental Studies Hybridoma Bank, Iowa City, IA, USA; BA-F8, SC-71, and BF-F3) and laminin (Abcam, Cambridge, MA, USA, 11575). Representative $\times$20 images were acquired from the soleus mid-belly, plantaris mid-belly, red gastrocnemius and white gastrocnemius. Fibre type and cross-sectional area were determined using a semi-automated ImageJ macro with unstained fibres classified as 2$\times$. Greater than 50 fibres of each type were included in the analyses.

MEF-derived adipose depots were fixed for >48 h in Pen-Fix (Richard Allan Scientific, Kalamazoo, MI, USA), embedded in paraffin, sectioned at 10 $\mu$m and stained with haematoxylin and eosin.

### Quantitative RT-PCR

RNA was isolated from the superficial (white) portion of the gastrocnemius muscle by Trizol–chloroform extraction of samples homogenized using a TissueLyser II (Qiagen, Hilden, Germany). RNA concentration and purity were assessed using a Nanodrop spectrophotometer (Thermo Fisher Scientific, Waltham, MA, USA) and 1 $\mu$g of total RNA was reverse transcribed using MultiScribe (Thermo Fisher Scientific). Expression of MAFbx/atrogin1 (*Fbxo32*: F: AACCGGGAGGCCAGCTAAAGAACA, R: TGGGCCTACAGAACAGACAGTGC), MuRF1 (*Trim63*: F: GAGAACCTGGAGAAGCAGCT, R: CCGCGGT TGGTCCAGTAG), myostatin (*Mstn*: F: CAGACCCGTC AAGACTCCTACA, R: CAGTGCCTGGGCTCATGTC AAG) and glyceraldehyde 3-phosphate dehydrogenase (*Gapdh*: F: TGTGATGGGTGTGAACCACGAGAA, R: GAGCCCTTCCACAATGCCAAAGTT) was assessed in duplicate by quantitative real time PCR using Fast SYBR Green and a QuantStudio3 (Thermo Fisher Scientific) with the manufacturer's settings for Fast SYBR. Relative expression of *Fbxo32*, *Trim63* and *Mstn* was calculated by the $\Delta\Delta C_\mathrm{t}$ method with normalization to *Gapdh*.

### Western blotting

Protein was extracted from frozen muscle in RIPA buffer supplemented with Complete Protease Inhibitor (Roche, Basel, Switzerland) with bead homogenization using a TissueLyser II (Qiagen). Homogenized tissue was then solubilized for 1 h at 4°C with agitation, centrifuged and the protein concentration of the supernatant determined by a Pierce BCA assay (Thermo Fisher Scientific) according to the manufacturer's instructions. Then, equivalent amounts of protein (40 $\mu$g) diluted in diH$_2$O with Laemmli buffer were denatured and separated on 4–12% Bis-Tris gels. Protein was then transferred to polyvinylidene difluoride membrane, reversibly stained with ponceau S and blocked in TBST+ (1$\times$ Tris-buffered saline with 2.5% fish gelatin, 0.1% sodium azide and 0.5% Tween). The following primary antibodies from Cell Signaling Technology (Danvers, MA, USA) were applied overnight at 4°C at 1:1000 unless otherwise noted: mTOR (Cell Signaling Technology; 2983), phospho-mTOR-ser2448 (Cell Signaling Technology; 5536), Akt (Cell Signaling Technology; 9272), phospho-Akt-ser473 (Cell Signaling Technology; 4060), S6 ribosomal protein (Cell Signaling Technology; 2317), phospho-S6 ribosomal protein-ser235/236 (Cell Signaling Technology; 4858), puromycin (EMD Millipore, Billerica, MA, USA; MABE343) and actin (Sigma, St Louis, MO, USA; A2066; 1:10,000). Membranes were washed and incubated with an appropriate secondary antibody and imaged with a LI-COR Odyssey (LI-COR, Lincoln, NE, USA). Blot analysis was performed using Image Studio (LI-COR). Band intensities were normalized to the total protein correlate of each phosphorylated signalling protein.

### Serum and conditioned medium assessments

Serum was collected from a retroorbital vein in a serum separator tube and allowed to clot at room temperature. Conditioned medium (CM) was generated from 0.250 g of tissue, which was incubated in PBS for 1 h followed by 24 h culture in low glucose (1%) Dulbecco's modified Eagle's medium in a 37°C incubator on an orbital shaker.

After 24 h, remaining adipose segments were removed and CM was snap frozen in liquid nitrogen. Both serum and CM were stored at $-80$°C until analysis by Luminex multiplex 31-plex chemokine/cytokine array assay (Eve Technologies, Calgary, AB, Canada). Serum and CM were assessed at a dilution of 1:2.

An enzyme-linked immunosorbent assay (ELISA) for serum leptin was conducted using a mouse/rat Leptin Quantikine ELISA Assay kit (MOB00; R&D Systems) at a dilution of 1:2 in FF mice, and all other groups were evaluated at the manufacturer's recommended 1:20 dilution. The lower limit of detection of the assay was 22 pg/ml. ELISA for corticosterone was conducted using a Corticosterone ELISA kit (K014-H1; Arbor Assays, Ann Arbor, MI, USA), which has a lower limit of detection to 17.5 pg/ml. Interassay variability for both kits was $<5\%$ between two runs of each plate.

Serum free fatty acid (FFA) levels were determined in a two-step reaction. First, 2 volumes of reagent A was added to all wells, 1 $\mu$l of plasma or standard added, and the mixture incubated for 15 min at room temperature. Then, colour was developed with the addition of 1 volume of reagent B to all wells. The plate with samples and standards was incubated for an additional 15 min at room temperature, read at 540 nm, and corrected for blanks and a secondary wavelength at 660 nm.

Serum triglyceride (TG) levels were calculated in a 96-well plate format, where 100 $\mu$l of each reagent is aliquoted. To the reagent, 1 $\mu$l of serum sample or standard was added and mixed thoroughly. The reaction was then incubated for 30 min at room temperature. Total triglyceride levels were determined by readings at 540 nm wavelength on a plate reader. Sample concentrations were calculated from a standard curve correcting for blanks and a secondary wavelength at 660 nm.

### Statistical analysis

An *a priori* power analysis indicated $n = 6$ per group to observe a statistical difference in EDL muscle tetanic force ($\alpha < 0.05$; $1 - \beta = 0.85$) based on previously published data (Biltz et al., 2020). Grouped data were compared using Student's *t*-test or one- or two-way analysis of variance (ANOVA) where appropriate with pairwise comparisons evaluated by Tukey's or Šidák's *post hoc* test. Serum and CM mediators were assessed first using a Brown–Forsythe test to evaluate equality of variances. If not significant, data were evaluated using one-way ANOVA with Tukey's multiple comparisons test. If the Brown–Forsythe test indicated unequal variance, data were instead compared using a Brown–Forsythe and Welch non-parametric one-way ANOVA with Dunnett's multiple comparisons test. All statistical analyses were performed in GraphPad Prism (GraphPad Software Inc., San Diego, CA, USA) except hierarchical clustering which was performed using the hclust algorithm in R (R Foundation for Statistical Computing, Vienna, Austria) with the complete linkage method. All data are represented as means ± standard deviation with numbers per group and exact statistical tests listed in the figure legends.

## Results

### Fat-free mice exhibit a muscle mass deficit driven by fast-fibre atrophy

Fat-free (FF) mice have smaller crown–rump length (Fig. 1*A*), a complete absence of adipose tissue depots (Fig. 1*B*), but similar body mass at both 6 and 16 weeks of age (Fig. 1*C*) when compared to WT littermate control mice, consistent with previous reports (Collins et al., 2020; Wu et al., 2018). Maintenance of body mass is due to an increase in liver size, which offsets the reduction in adipose mass in WT mice (Collins et al., 2020). FF mice have 20–30% lower mass of predominantly fast-fibred muscles apparent by gross inspection (Fig. 1*D*; top, gastrocnemius), with qualitatively similar mass of slow-fibred muscles (Fig. 1*D*; bottom, soleus). The mass deficit in the FF fast-fibred gastrocnemius and plantaris muscles is significant as early as 6 weeks of age and is maintained through adulthood (Fig. 1*E*). By contrast, the mass of the slow-fibred soleus was not significantly different between WT and FF groups at 6 or 16 weeks of age (Fig. 1*E*). However, two-way ANOVA found a main effect of genotype owing to the consistent ∼10% mass deficit in the FF group. This suggests the FF muscle pathology is exacerbated in fast twitch muscles. To quantify this specificity, we evaluated the cross-sectional area (CSA) of the four fibre types in the plantarflexor muscle group (gastrocnemius–plantaris–soleus). Type 2b were the only fibres with lower CSA in FF compared with WT muscle, approaching significance ($P = 0.06$) at 6 weeks of age with a significant reduction by 16 weeks (Fig. 1*F*). Type 2a fibre area was significantly increased in FF mice at 16 weeks, but the increase was not present at 6 weeks. Type 1 and type 2x fibre types were not different between FF and WT muscles. The relative distribution of fibre types was only different in the soleus muscle of FF mice at 16 weeks of age, which exhibited a significant increase in type 1 and a significant decrease in type 2x fibres (Fig. 1*G*). These data indicate that adipose ablation causes a muscle mass deficit due to specific atrophy of type 2b fibres.

### Mouse embryonic fibroblast treatment rescues the FF mass and contractile deficit

To determine if restoration of adipose tissue could reverse the muscle mass deficit in FF mice, a group

of 6-week-old FF mice were implanted with mouse embryonic fibroblasts (MEFs) isolated from WT mice, which have been previously shown to generate a spontaneous fat pad of ~1 g mass (Collins et al., 2020) when injected subcutaneously into the sternal region of

FF mice. The injectate was retained in the subcutaneous space superficial to the sternal muscles and later, upon explantation, fat pads exhibited a white adipose tissue-like morphology (Fig. 2A). At 16 weeks, MEF treatment of FF mice rescued gastrocnemius and plantaris mass to

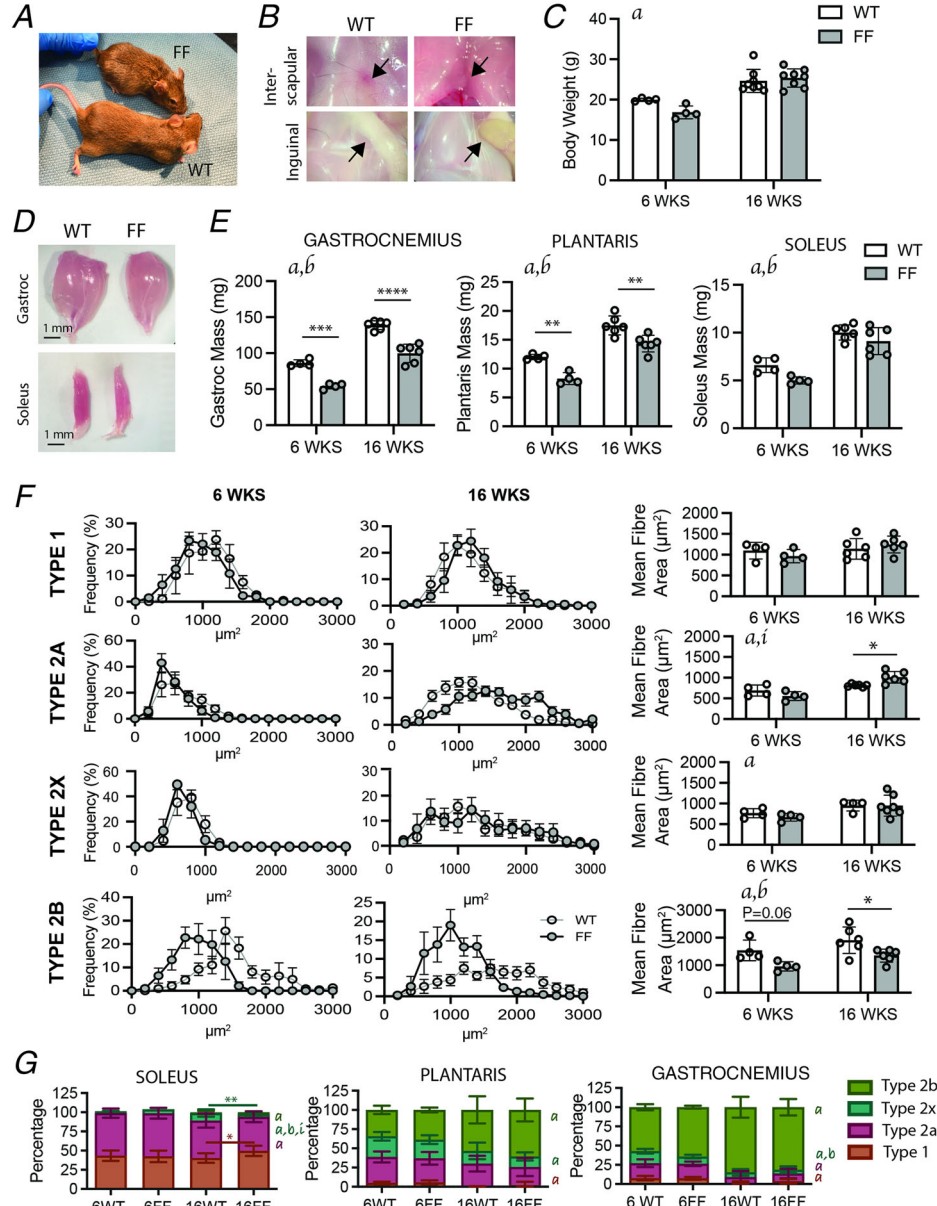

**Figure 1. Fat-free mice exhibit a muscle mass deficit driven by fast-fibre atrophy**

*A*, image of Fat-free (FF) and wild-type littermate control mice (WT), illustrating smaller length in FF mice. *B*, FF mice lack both brown (interscapular; top arrows) and white (subcutaneous; bottom arrows) adipose depots. *C*, FF mice have similar body mass to WT mice at 6 weeks and 16 weeks of age. *D*, FF fast-fibred muscles (gastrocnemius; top) are smaller than WT by gross inspection while slow-fibred muscles (soleus; bottom) are similar in size. *E*, mass of gastrocnemius and plantaris muscles are smaller in FF at 6 weeks and 16 weeks, while soleus muscles are unaffected. *F*, binned frequency curves and mean of fibre cross-sectional area for each muscle fibre type averaged over the plantarflexor muscle complex. Only type 2b fibres exhibit a leftward shift and lower mean in the FF group. *G*, only the soleus muscle had significant changes in the relative percentages of each fibre type in the FF group at 16 weeks. Data compared by two-way ANOVA (*a*: main effect of age, *b*: main effect of genotype, *i*: age–genotype interaction) with Šidák's *post hoc* test (*$P < 0.05$, **$P < 0.01$, ***$P < 0.005$, ****$P < 0.001$).

WT levels (Fig. 2*B*), and reversed type 2b fibre atrophy observed in FF mice (Fig. 2*C*), confirming the pathology derives from a lack of adipose and retains plasticity in response to adipose-derived factors. Type 2a fibre area was unaffected by MEF treatment and also unaffected by FF in this dataset (Fig. 2*C*). Consistent with the lack of phenotype in type 1 and 2x fibre in FF muscle, MEF treatment did not affect these fibre types.

To explore the functional consequences of this pathology, contractile testing was performed on a pre-dominantly fast extensor digitorum longus (EDL) and predominantly slow (soleus) muscle. MEF treatment also reversed the loss of mass in the EDL muscle of FF mice but did not significantly alter soleus muscle mass (Fig. 2*D*). Specific peak tetanic tension was 20% lower in the FF EDL, but 15% higher in the FF soleus compared

with WT (Fig 2*E*). Similar results were observed at 6 weeks of age (Table 1). As this tension is normalized to PCSA, accounting for changes in muscle mass, this suggests an intrinsic force production deficit specifically in fast-fibred muscles of FF mice. Not only did MEF treatment rescue EDL mass, it also restored EDL peak tetanic tension to WT levels (Fig. 2*E*). Twitch tension and time to peak tension were not altered (Table 1), but there was a significant increase in the half-relaxation time in the FF EDL (Fig. 2*F*) suggesting a delay in calcium uptake in the sarcoplasmic reticulum or altered rates of cross-bridge detachment. This effect was also observed at 6 weeks of age in both muscles (Table 1). The increased half-relaxation time was not fully rescued by MEF treatment. Consistent with the lack of fibre type changes observed, there were no differences in fatigability in either muscle between

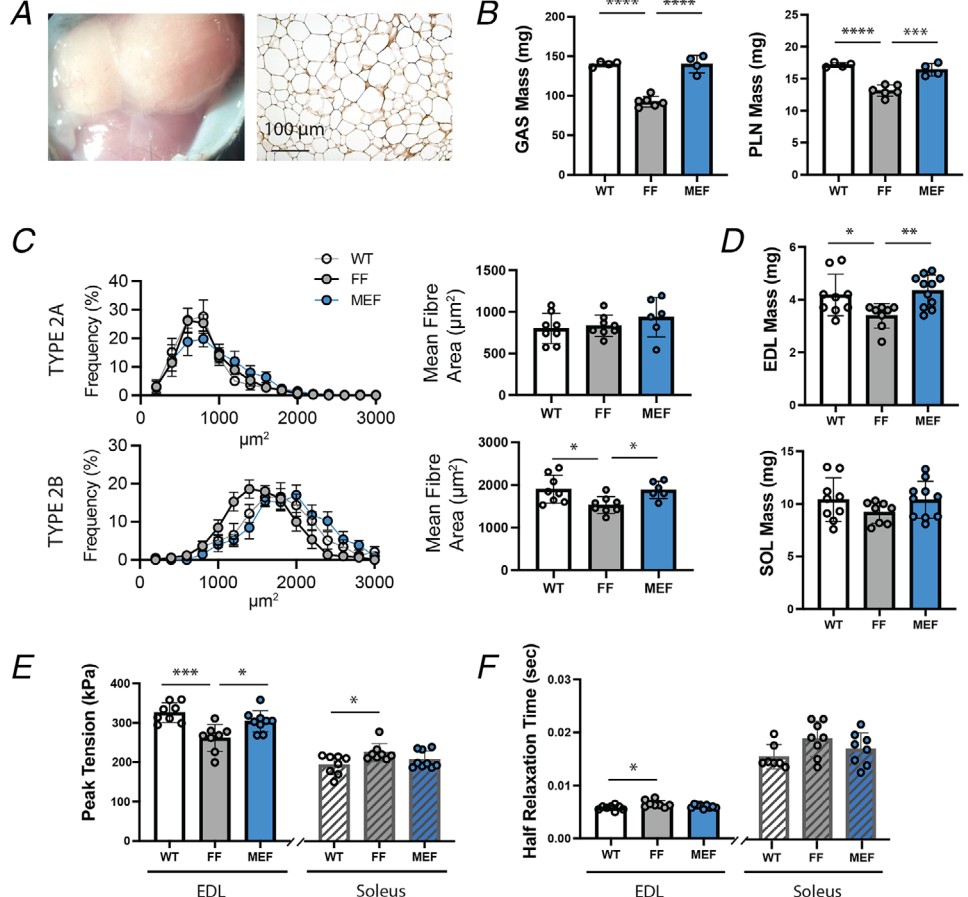

**Figure 2. MEF transplant rescues muscle mass and contractile deficit in FF mice**
*A*, gross morphology and midsection histology of MEF fat pad resemble white adipose. *B*, MEF treatment (blue bars) fully rescues gastrocnemius (GAS) and plantaris (PLN) mass to WT (white bars) values. *C*, frequency distributions and means of type 2a and 2b fibres in the plantarflexor muscle complex. MEF treatment rescues the leftward shift and lower mean in type 2b fibres. *D*, MEF treatment rescues the mass deficit in the FF fast-fibred extensor digitorum longus (EDL), and does not affect mass of the slow-fibred soleus (SOL). *E*, MEF treatment fully rescues the deficit in peak tetanic tension in FF EDL muscles (filled bars), but does not significantly affect the potentiation of peak tension in the FF soleus (hatched bars). *F*, MEF treatment does not affect the twitch force relaxation (half-relaxation time) in either the EDL or soleus muscle. Full contractile data are reported in Table 1. Data compared with one-way ANOVA with Tukey's *post hoc* test (*$P < 0.05$, **$P < 0.01$, ***$P < 0.005$, ****$P < 0.001$).

**Table 1. Contractile data for 6 and 16 weeks for wild-type, fat-free and MEF-corrected mice**

| Outcome | 6 weeks | | | 16 weeks | | | P (one-way ANOVA main effect) |
|---|---|---|---|---|---|---|---|
| | WT (n = 4) | FF (n = 4) | P (t-test) | WT (n = 9) | FF (n = 8) | MEF (n = 9) | |
| **EDL** | | | | | | | |
| FL (mm) | 5.82 ± 0.40 | 6.22 ± 0.34 | 0.176 | 6.66 ± 0.40 | 6.58 ± 0.57 | 7.18 ± 0.45 | 0.27 |
| MW (mg) | 2.85 ± 0.21 | 2.28 ± 0.28 | 0.016 | 4.05 ± 0.66 | 3.43 ± 0.49 | 4.70 ± 1.14 | 0.01 |
| PCSA (mm$^2$) | 0.45 ± 0.04 | 0.33 ± 0.05 | 0.016 | 0.55 ± 0.07 | 0.48 ± 0.07 | 0.60 ± 0.15 | 0.06 |
| Twitch tension (kPa) | 40.58 ± 6.22 | 40.47 ± 9.76 | 0.986 | 53.87 ± 8.98 | 54.10 ± 8.22 | 48.87 ± 16.07 | 0.52 |
| Time to peak tension (ms) | 0.0052 ± 0.0006 | 0.0065 ± 0.0010 | 0.067 | 0.0052 ± 0.0005 | 0.0054 ± 0.0004 | 0.0059 ± 0.0006 | 0.02 |
| Half-relaxation time (ms) | 0.0063 ± 0.0005 | 0.0080 ± 0.0014 | 0.058 | 0.0058 ± 0.0005 | 0.0065 ± 0.0006 | 0.0063 ± 0.0004 | 0.03 |
| Peak tetanic tension (kPa) | 256.48 ± 15.68 | 219.36 ± 25.79 | 0.049 | 324.18 ± 24.45 | 261.45 ± 34.41 | 303.78 ± 26.79 | <0.001 |
| Time to fatigue (50%) | 175.66 ± 25.08 | 248.87 ± 83.59 | 0.144 | 164.51 ± 42.49 | 157.84 ± 52.93 | 170.71 ± 55.49 | 0.88 |
| **Soleus** | | | | | | | |
| FL (mm) | 7.26 ± 0.40 | 6.80 ± 0.33 | 0.125 | 6.93 ± 0.73 | 6.81 ± 0.87 | 7.26 ± 0.32 | 0.54 |
| MW (mg) | 6.60 ± 0.76 | 5.00 ± 0.37 | 0.009 | 10.18 ± 1.78 | 8.20 ± 1.03 | 10.41 ± 1.77 | 0.27 |
| PCSA (mm$^2$) | 0.83 ± 0.13 | 0.67 ± 0.04 | 0.052 | 1.36 ± 0.34 | 1.22 ± 0.13 | 1.31 ± 0.25 | 0.37 |
| Twitch tension (kPa) | 22.37 ± 2.07 | 29.12 ± 1.88 | 0.003 | 19.14 ± .99 | 21.52 ± 4.13 | 21.88 ± 4.01 | 0.46 |
| Time to peak tension (ms) | 0.0122 ± 0.0010 | 0.0155 ± 0.0032 | 0.101 | 0.0099 ± 0.0015 | 0.0112 ± 0.0015 | 0.0114 ± 0.0019 | 0.13 |
| Half-relaxation time (ms) | 0.0183 ± 0.0015 | 0.0230 ± 0.0022 | 0.011 | 0.0157 ± 0.0023 | 0.0191 ± 0.0034 | 0.0180 ± 0.0037 | 0.14 |
| Peak tetanic tension (kPa) | 196.24 ± 26.93 | 212.78 ± 15.24 | 0.326 | 190.65 ± 24.72 | 219.41 ± 22.69 | 206.68 ± 22.21 | 0.04 |
| Time to fatigue (50%) | 597.85 ± 77.85 | 594.25 ± 100.54 | 0.957 | 339.36 ± 164.60 | 328.59 ± 83.46 | 284.24 ± 82.14 | 0.62 |

Values are means ± SD. FL, fiber length; MW, muscle weight; PCSA, physiological cross-sectional area.

groups (Table 1). Taken together these data suggest that absence of adipose causes a complex pathology, specifically in fast twitch muscle fibres, including lower mass, a deficit in force-producing capacity and altered contractile dynamics. As fat is an active endocrine organ, we next sought to determine which secreted factors may be governing the rescue of muscle mass and physiology by the MEF fat pad and, by extension, the factors from native fat that regulate these processes.

### MEF fat pad resembles visceral adipose and secretes a number of factors that can impact muscle mass

We began with a targeted screen of MEF explant conditioned medium (CM) using a Luminex 31-plex assay. Consistent with the 'white' adipose appearance of MEF adipocytes (Fig. 2*A*), the cytokine secretion of MEF explants most resembled that of epididymal visceral adipose – the classic 'white' adipose depot in mice. The majority of MEF CM samples clustered with visceral adipose CM, while brown fat CM clustered independently (Fig. 3*A*). MEF explants secreted high levels of several visceral-associated inflammatory cytokines thought to be important for the negative adipose–muscle signalling in obesity, notably interleukin 6 (IL-6), monocyte chemoattractant protein 1 (MCP-1), tumour necrosis factor α (TNF-α) and IL-1β (Deng & Scherer, 2010) (Fig. 3*B*). However, the multiplex assay indicated that MEF explants also secrete factors thought to promote muscle hypertrophy, including leukaemia inhibitor factor (LIF) and IL-15 (Fig. 3*C*). We also assessed two additional adipokines that impact muscle physiology which were not included in the multiplex: adiponectin and leptin (Fig. 3*D*). Secretion of adiponectin was higher in the MEF explant than subcutaneous or visceral depots, but was on a par with interscapular brown fat. By contrast, leptin secretion was about 4-fold higher in the MEF explant than all the other adipose depots. Taken together, this suggests that the endocrine profile of MEF adipose resembles native adipose and that despite its small volume, MEF adipose could restore a portion of native adipose–muscle signalling. Given the hypothesized anabolic role for leptin on muscle and the similarity in muscle phenotype between the FF and leptin knockout (*ob*/*ob*) mouse we first investigated whether restoration of leptin signalling could underlie the phenotypic rescue with MEF treatment.

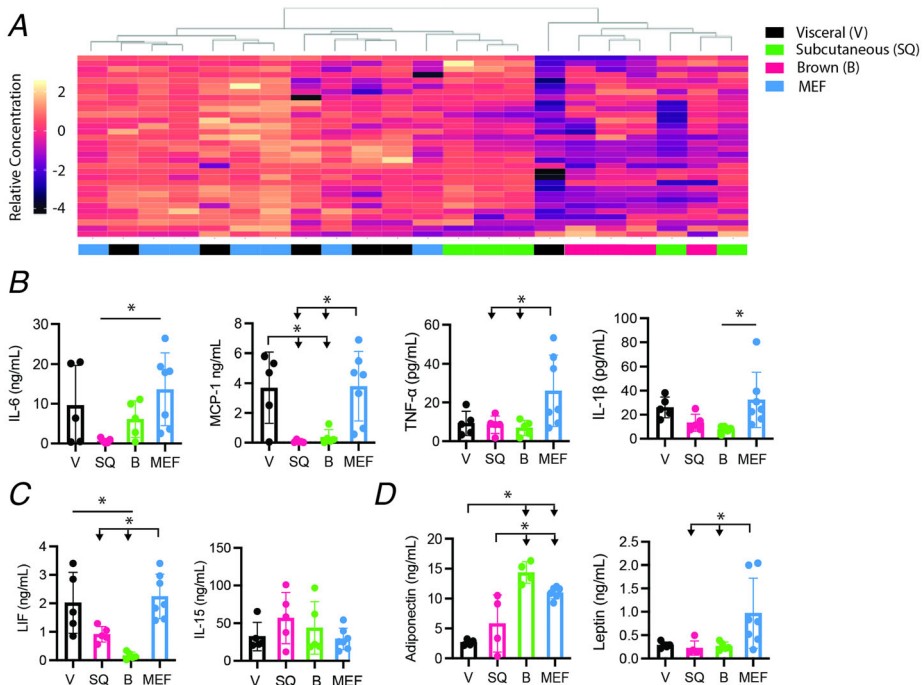

**Figure 3. MEF explant is most similar to WAT and secretes a number of factors likely to impact muscle mass**

*A*, heatmap of relative concentrations of cytokine, adipokine and growth factor mediators detected in conditioned medium (CM) from visceral (V, black), subcutaneous (SQ, green), brown (B, pink) and MEF (blue) tissues by multiplex array predominantly clusters MEF CM with V CM. *B*, MEF explant secretes high levels of classical inflammatory adipokines interleukin (IL)-6, macrophage chemoattractant protein-1 (MCP-1), tumour necrosis factor α (TNF-α) and IL-1β. *C*, MEF explant also secretes muscle anabolic factors leukaemia inhibitory factor (LIF) and IL-15. *D*, MEF explant secretes high levels of classic adipokines adiponectin and leptin, assessed by ELISA. Data compared with 1-way ANOVA with Tukey's *post hoc* test, \*P < 0.05.

### Leptin knockout MEF treatment fails to rescue muscle phenotype

To separate the effect of leptin from recovery of other signalling pathways, leptin knockout MEFs (MEF-OB) were generated and implanted into FF mice. MEF-OB fat pads were of similar size and gross appearance to those from standard MEF treatment (Fig. 4*A*). While MEF treatment restored a range of circulating leptin from ∼5% to 50%, MEF-OB treatment resulted in detectable levels of circulating leptin in only two mice (0.65 and 0.23 ng/ml, respectively), which is likely due to non-specific binding, and virtually undetectable levels for leptin were observed in MEF-OB explant CM for all mice (Fig. 4*B*). Gastrocnemius, plantaris and EDL muscle mass was

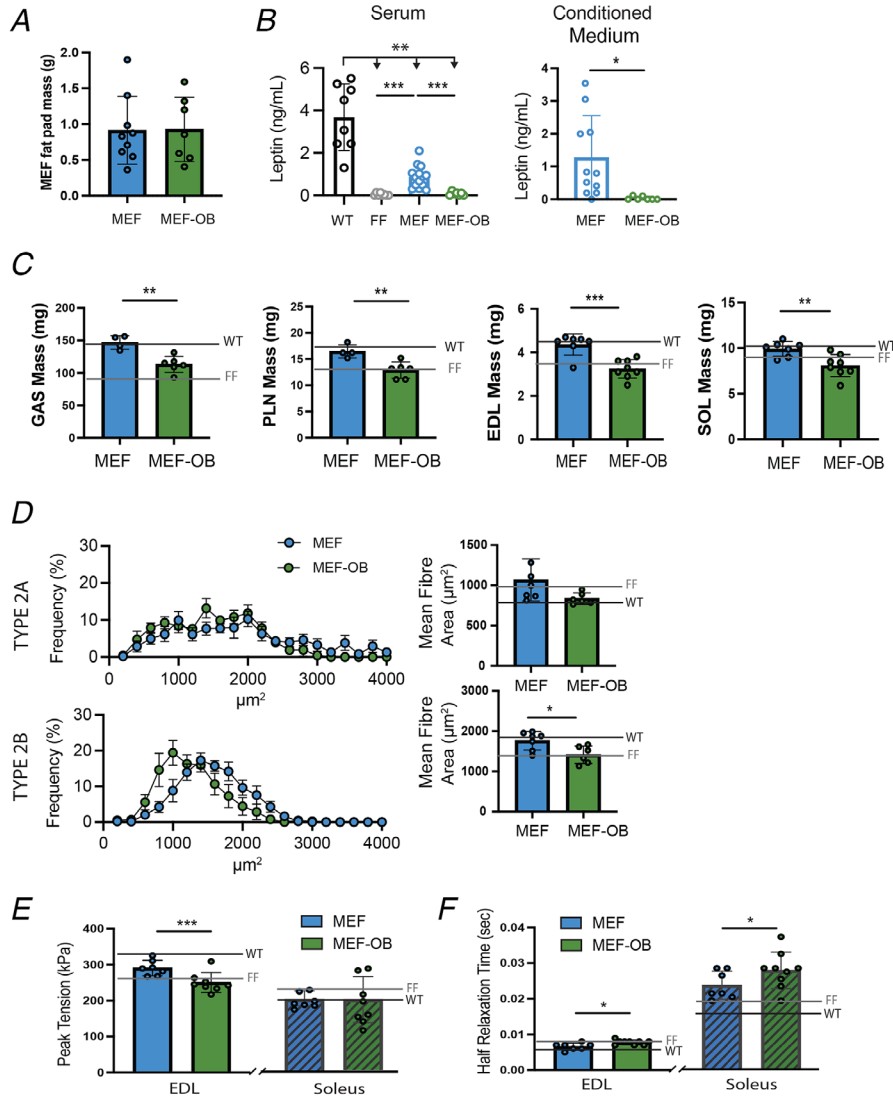

**Figure 4. Leptin deficient MEFs fail to rescue the muscle phenotype**
*A*, MEF-OB treatment creates a fat pad of similar size to standard MEF. *B*, MEF-OB treated mice (green bars) have significantly lower serum leptin compared with standard MEF treatment (blue bars), and comparable serum leptin to FF (grey). MEF-OB conditioned medium also has nearly undetectable levels of leptin. *C*, gastrocnemius (GAS), plantaris (PLN), EDL and soleus (SOL) mass is significantly lower with MEF-OB compared with standard MEF treatment. *D*, MEF-OB treated FF mice have a leftward shift of the type 2b fibre area distribution curve and significantly lower mean area of type 2b fibres in the plantarflexor muscle group. Type 2a fibre area is not significantly affected by MEF type. *E*, MEF-OB treated mice have significantly lower peak tetanic tension in the EDL (filled bars) compared with standard MEF. Soleus peak tetanic tension (hatched bars) is not significantly affected by MEF type. *F*, MEF-OB treatment causes a significant prolongation of twitch relaxation (half-relaxation time) in both EDL and soleus muscles. Discrepancy in muscle mass, type 2b fibre areas, peak tetanic tension and half-relaxation time between MEF and MEF-OB are consistent with a WT to FF comparison which is noted by black and grey lines on each plot, respectively. Data compared by unpaired *t*-test, *$P < 0.05$, **$P < 0.01$, ***$P < 0.005$.

significantly lower in MEF-OB treated FF mice compared to MEF-WT (Fig. 4*C*) and both masses were similar to untreated FF (Fig. 4*B*; grey lines). Interestingly, the mass of the soleus muscle was also significantly lower in MEF-OB treated FF mice. Type 2b fibre cross-sectional area was also significantly lower with MEF-OB treatment, while type 2a fibre area was not significantly different between the two MEF correction groups (Fig. 4*D*). EDL peak tetanic tension was significantly lower in MEF-OB treated mice at a level comparable to untreated FF (Fig. 4*E*). However, soleus muscles from MEF-OB treated mice did not exhibit the increased peak tetanic tension of untreated FF. Similar to FF, muscles from MEF-OB treated mice had an increased half-relaxation time which was significant for both the EDL and soleus (Fig. 4*F*). Thus, reconstitution of adipose tissue with MEFs lacking leptin failed to recapitulate the rescue achieved by WT MEFs, indicating that the prevention of muscle pathology in MEF-corrected FF mice was likely mediated by leptin.

### The lack of efficacy of MEF-OB treatment is not due to dysregulated adipokines

Adipose tissue from *ob/ob* mice has an altered profile of secreted adipokines notably characterized by dramatically elevated TNF-$\alpha$ and reduced adiponectin (Favero et al., 2015), both of which would be expected to negatively impact muscle mass. To investigate whether the lack of efficacy of MEF-OB treatment was due to the lack of leptin secretion or differential secretion of other adipokines, we repeated the multiplex adipokine screen to compare MEF and MEF-OB CM. MEF-OB CM samples did not fully cluster separately from MEF CM, though an exclusive MEF-OB and an exclusive MEF cluster were identified (Fig. 5*A*). However, the differences between MEF and MEF-OB were subtle and only IL-3 was significantly different between groups (Fig. 5*B*). Importantly, the concentrations of adipokines most likely to reduce muscle mass based on the literature were not different between groups (Fig. 5*C*). The anabolic cytokines LIF and IL-15 were also not different (Fig. 5*D*); however, secretion of adiponectin (assessed by ELISA) was significantly higher in MEF explants compared with MEF-OB, consistent with findings in native adipose from *ob/ob* mice (Fig. 5*E*). With MEF treatment, this difference was not sustained at the serum level, however, as MEF-OB treated mice had slightly higher circulating adiponectin compared with MEF treated mice (Fig. 5*F*). In summary, the absence of leptin did not substantially alter the MEF-secreted adipokine profile.

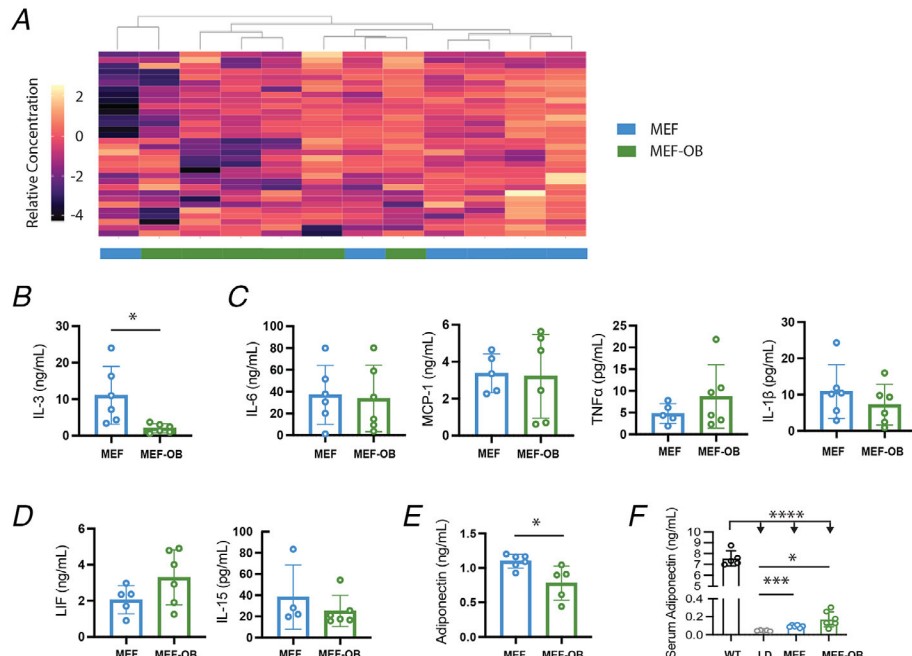

**Figure 5. MEF-OB does not secrete dramatically dysregulated adipokines compared with standard MEF**
*A*, hierarchical clustering on 31 cytokines in a multiplex array fails to separately cluster MEF and MEF-OB conditioned medium (CM). Similarities in secretion profiles between MEF and MEF-OB are apparent by visual inspection of the heatmap. *B*, interleukin (IL)-3 secretion is higher in MEF than MEF-OB CM. *C*, MEF and MEF-OB CM have similar levels of inflammatory adipokines with catabolic action on muscle: interleukin (IL)-6, macrophage chemoattractant protein-1 (MCP-1), tumour necrosis factor $\alpha$ (TNF-$\alpha$) and IL-1$\beta$. *D*, MEF and MEF-OB CM have similar levels of anabolic cytokines leukaemia inhibitory factor (LIF) and IL-15. *E* and *F*, MEF-OB CM has significantly lower levels of adiponectin (*E*), but this fails to significantly impact serum adiponectin (*F*). *$P < 0.05$, ***$P < 0.005$, ****$P < 0.001$.

### MEF treatment rescue is not mediated by reversal of systemic metabolic dysfunction

As leptin is a pleiotropic adipokine that is known to impact other physiological systems, we sought to explore the mechanism by which adipose-secreted leptin mediates the regulation of muscle mass and strength. Leptin has a potent effect on insulin signalling and treatment of *ob/ob* and other lipodystrophic mice with physiological leptin doses rescues insulin resistance (Ebihara et al., 2001; Wendel et al., 2008). However, we find that MEF-secreted leptin is insufficient to fully rescue insulin resistance in

FF mice, measured by the AUC of the fasting insulin tolerance test (Fig. 6*A*). Furthermore, insulin resistance was similarly mitigated with both MEF and MEF-OB treatment, suggesting it is not sufficient to mediate the effect of MEF rescue. Glucose tolerance was similar between WT, FF and MEF but significantly increased in MEF-OB when compared with MEF (Fig. 6*A*). Serum triglycerides and free fatty acids were equally normalized by both MEF and MEF-OB treatments (Fig. 6*B*). Elevated glucocorticoids also directly contribute to muscle atrophy (Menconi et al., 2008), and reduction in circulating

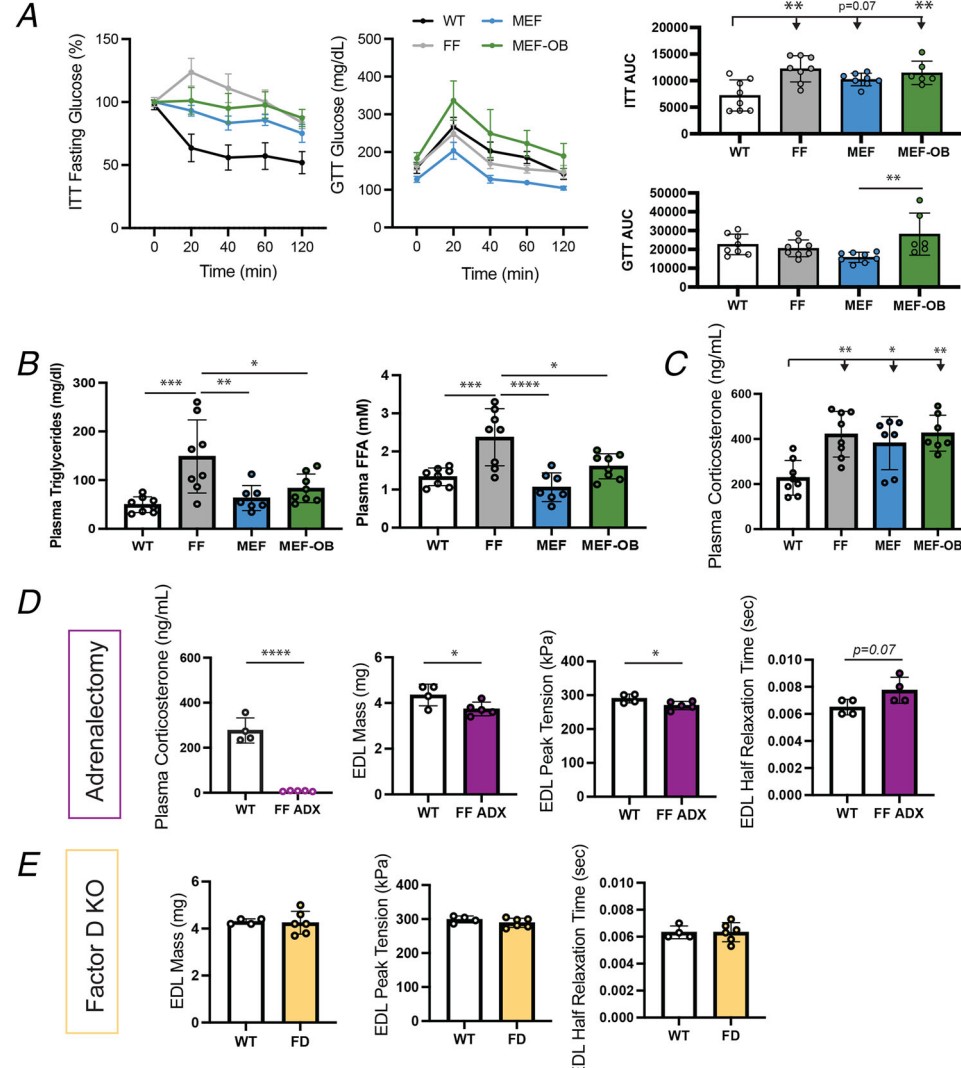

**Figure 6. Leptin does not mediate the effect of MEF treatment on metabolic dysfunction**
*A*, insulin tolerance test (ITT) glucose as a percentage of fasting glucose, glucose tolerance test (GTT) glucose, and area under the curve (AUC) calculations for both outcomes measured for WT (black), FF (grey), MEF (blue), and MEF-OB (green). Both MEF and MEF-OB fail to fully rescue the elevated ITT AUC in FF mice. *B*, both MEF and MEF-OB normalize elevated serum triglycerides and free fatty acid (FFA) levels of FF to WT. *C*, elevated plasma corticosterone in FF mice is not affected by either MEF treatment. *E*, adrenalectomy dramatically reduces circulating corticosterone, but fails to rescue EDL mass, peak tetanic tension or prolongation of twitch relaxation. *F*, knockout of adipsin/factor D (FD) does not cause a deficit in EDL mass, peak tetanic tension or prolongation of twitch relaxation. \*$P < 0.05$, \*\*$P < 0.01$, \*\*\*$P < 0.005$, \*\*\*\*$P < 0.001$.

glucocorticoids via bilateral adrenalectomy in *ob/ob* mice improves muscle mass (Smith & Romsos, 1985) suggesting a potential adipose–adrenal–muscle axis mediated by leptin. Neither MEF nor MEF-OB treatment normalized the elevated plasma corticosterone in FF mice (Fig. 6*C*). Furthermore, adrenalectomy of FF mice, which dramatically reduced circulating corticosterone, failed to improve muscle mass or peak tetanic tension in FF mice (Fig. 6*D*). Elevated glucocorticoids are also thought to suppress secretion of adipsin/complement factor D from adipose tissue (Spiegelman et al., 1989) and so could also play a role in MEF treated FF mice. However, FD knockout mice do not exhibit the FF muscle mass or contractile phenotype, indicating FD secreted by fat is not involved in the onset or reversal of this pathology (Fig. 6*F*). Thus, the improvements in muscle mass and contractile function by MEF-secreted leptin are not mediated by reversal of the most likely systemic metabolic factors – insulin resistance, hyperlipidaemia and glucocorticoids.

Excessive ectopic accumulation of lipid in skeletal muscle and liver has been implicated in the development of anabolic resistance in diabetes (Meex et al., 2019) – specifically that lipotoxicity prevents normal loading cues from being translated into mass and strength gains. Both MEF and MEF-OB treatment decreased intramyocellular lipid (Fig. 7*A*) and liver mass (Fig. 7*B*) in FF mice. However, while MEF treatment nearly fully normalized values to WT levels, MEF-OB only partially reduced them. Importantly, there was a significant difference in liver mass between MEF and MEF-OB groups suggesting that leptin likely mediated part of the recovery in the MEF group. Similarly, MEF treatment fully rescued the deficit in tibia length in FF mice (Zhang et al., 2021), but MEF-OB treatment failed to do so (Fig. 7*C*). This suggests that the mechanism of MEF rescue of the FF muscle phenotype may not be through reversal of the typical systemic factors, but may still be mediated by a recovery of liver or skeletal health. MEF treatment delivered a range of circulating leptin in FF mice that was significantly correlated with

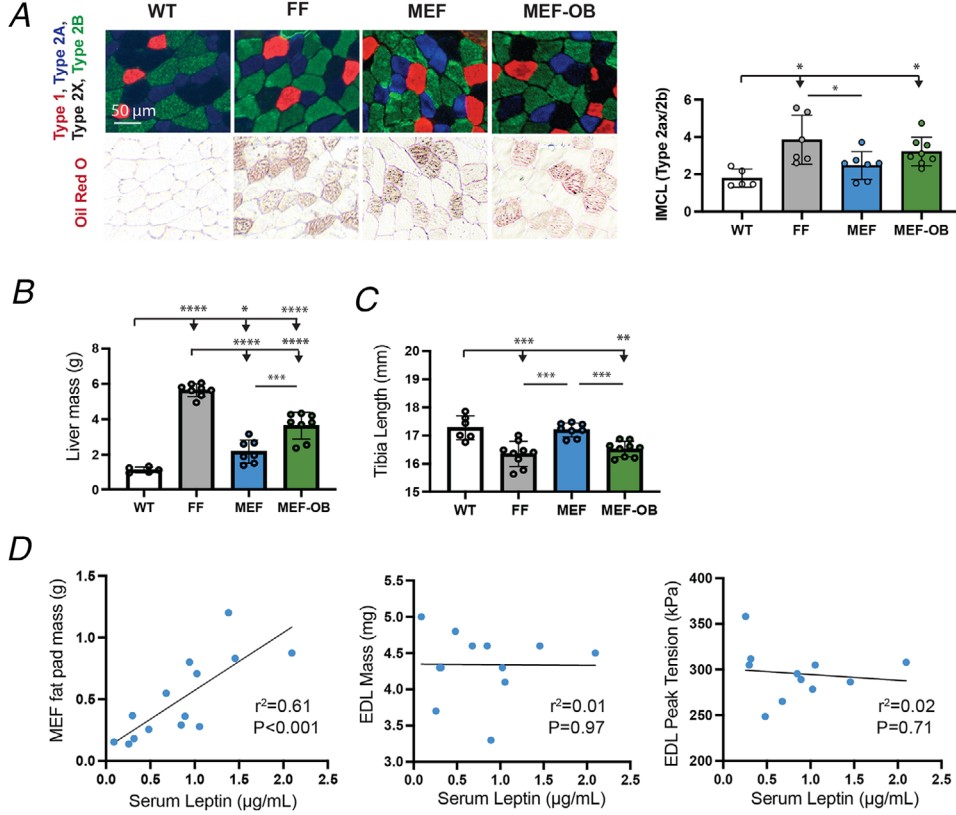

**Figure 7. Leptin mediates the effect of MEF treatment on other target tissues**
*A*, both MEF and MEF-OB partially reduce the FF elevated intramyocellular lipid (IMCL) accumulation in type 2a and 2x fibres. Qualitatively assessed by visual inspection (left) and quantitated as integrated density of Oil Red O in each fibre type. *B*, elevated liver mass in FF mice is more effectively rescued by standard MEF treatment than MEF-OB. *C*, tibia length deficits in FF mice are more effectively rescued by standard MEF treatment than MEF-OB. *D*, MEF fat pad mass (left), but not EDL mass (centre) or EDL peak tetanic tension (right), is significantly correlated with serum leptin. Grouped data compared with one-way ANOVA with Tukey's *post hoc* test, *x–y* data analysed by simple linear regression, \**P* < 0.05, \*\**P* < 0.01, \*\*\**P* < 0.005, \*\*\*\**P* < 0.001.

MEF fat pad mass (Fig. 7*C*). However, circulating leptin did not correlate with either EDL mass or peak tetanic tension (Fig. 7*C*). In fact, animals in the lowest quartile for circulating leptin (40% below the mean) had average rescue of EDL muscle mass and peak tension (5% below the mean), suggesting that the rescue of these parameters by MEF treatment was not dose dependent. Therefore, a minimum amount of adipose-derived leptin (0.09 ng/ml; 2.4% of WT average) is required to maintain muscle physiology and can be uncoupled from other factors previously thought to contribute to leptin's influence on muscle.

### Leptin promotes muscle protein synthesis without affecting atrogene-mediated degradation

Treatment of FF mice with exogenous leptin for 7 days resulted in a partial rescue of EDL and gastrocnemius mass (Fig. 8*A*). This short treatment time course was chosen to assess signalling mediators of hypertrophy and atrophy at a time when the muscle was actively adding mass. At this time point, there were no changes in expression of the ubiquitin ligase atrogenes MAFbx/atrogin-1 (*Fbxo32*) or MuRF1 (*Trim63*), which play a prominent role in other models of muscle atrophy (Fig. 8*B* red *vs*. grey). Interestingly, there were no differences in expression between saline treated FF mice (SAL) and WT, which was confirmed with additional samples from untreated FF mice (Fig. 8*B* white *vs*. grey). We did find a significant ∼2-fold upregulation of myostatin, an inhibitor of muscle growth, in both sample sets, but this was unchanged by 7 days of leptin treatment (Fig. 8*B*). As acute leptin treatment has been shown to induce phosphorylation of Akt in muscle (Maroni et al., 2003; Roman et al., 2010), we next investigated whether mTOR-driven protein synthesis could be mediating mass gains. We found a significant increase in phospho-Akt in leptin treated compared with saline treated gastrocnemius muscle and trends toward increased phospho-mTOR and phospho-S6 ribosomal protein (Fig. 8*C*). To determine whether these changes were driving increased protein synthesis, we blotted for puromycin to detect newly synthesized proteins by the SuNSET method (Goodman & Hornberger, 2013). Leptin treated muscles had significantly more puromycin incorporation (Fig. 8*D*) and in combination with the signalling data, this suggests that mass gains are mediated at least in part by mTOR-driven protein synthesis.

We next attempted to replicate these signalling changes with acute *ex vivo* leptin exposure to demonstrate direct action of leptin on muscle. Parallel incubation of EDLs from FF mice in Ringer's solution and Ringer's solution plus leptin elicited no differential effect in phosphorylation of Akt, mTOR, or S6 ribosomal protein

(Fig. 8*E*). In addition to hypertrophic signalling, we also assessed the ability of leptin incubation to modulate the prolongation of twitch half-relaxation time. *In vivo* leptin treatment fully rescued the prolonged twitch relaxation (Fig. 8*F*), but *ex vivo* incubation with leptin failed to do so (Fig. 8*G*). Together this points to an indirect mechanism for leptin's regulation of muscle mass and contractility.

## Discussion

Fat-free lipodystrophic mice that have a complete lack of adipose tissue exhibit a pronounced skeletal muscle phenotype characterized by decreased muscle mass and strength. Full rescue of the phenotype was achieved with replacement of just ∼10% of normal adipose mass, and furthermore, we demonstrated that this rescue is dependent on adipose-secreted leptin and separable from the reversal of systemic metabolic derangement. This work makes several advances in our understanding of adipose–muscle signalling. First, these data demonstrate that adipose tissue is required to achieve normal skeletal muscle mass and contractile performance. Second, we demonstrate that very little adipose is required for the maintenance of normal muscle mass and contraction. Reconstitution of less than 10% of WT adipose tissue fully rescued the FF muscle phenotype. Third, we show that adipose-secreted leptin specifically mediates the regulation of muscle mass and contraction. Though leptin has long been considered a major cytokine mediating adipose–muscle signalling, its role in regulating muscle physiology has not been well defined. We show that no other source of leptin (e.g. autocrine action of muscle-derived leptin on muscle) is sufficient to compensate for adipose-secreted leptin, no other adipokine is sufficient to compensate for leptin and that levels of circulating leptin far below normal are sufficient to normalize muscle physiology.

A number of partial lipodystrophic mouse models have been developed and described in the literature (reviewed in Rochford, 2014). These are either spontaneous mutations or models of gene mutations that give rise to lipodystrophy in humans and mimic those phenotypes including a gradient of adipose loss. Our strategy, which is a targeted genetic ablation of adipocytes, or cells that are adiponectin positive, is the only model to our knowledge that completely lacks adipose depots including the so called 'mechanical adipose', such as the fat pads of the paws. Interestingly, the muscular phenotype that we describe here is unique to this mouse among lipodystrophic models. Across mouse models where lean mass has been reported, none exhibit a deficit (Chen et al., 2012; Cortes et al., 2009; Guo et al., 2012). This aligns with lipodystrophy in humans which is not typically characterized by a loss of lean mass (Akinci et al., 2017;

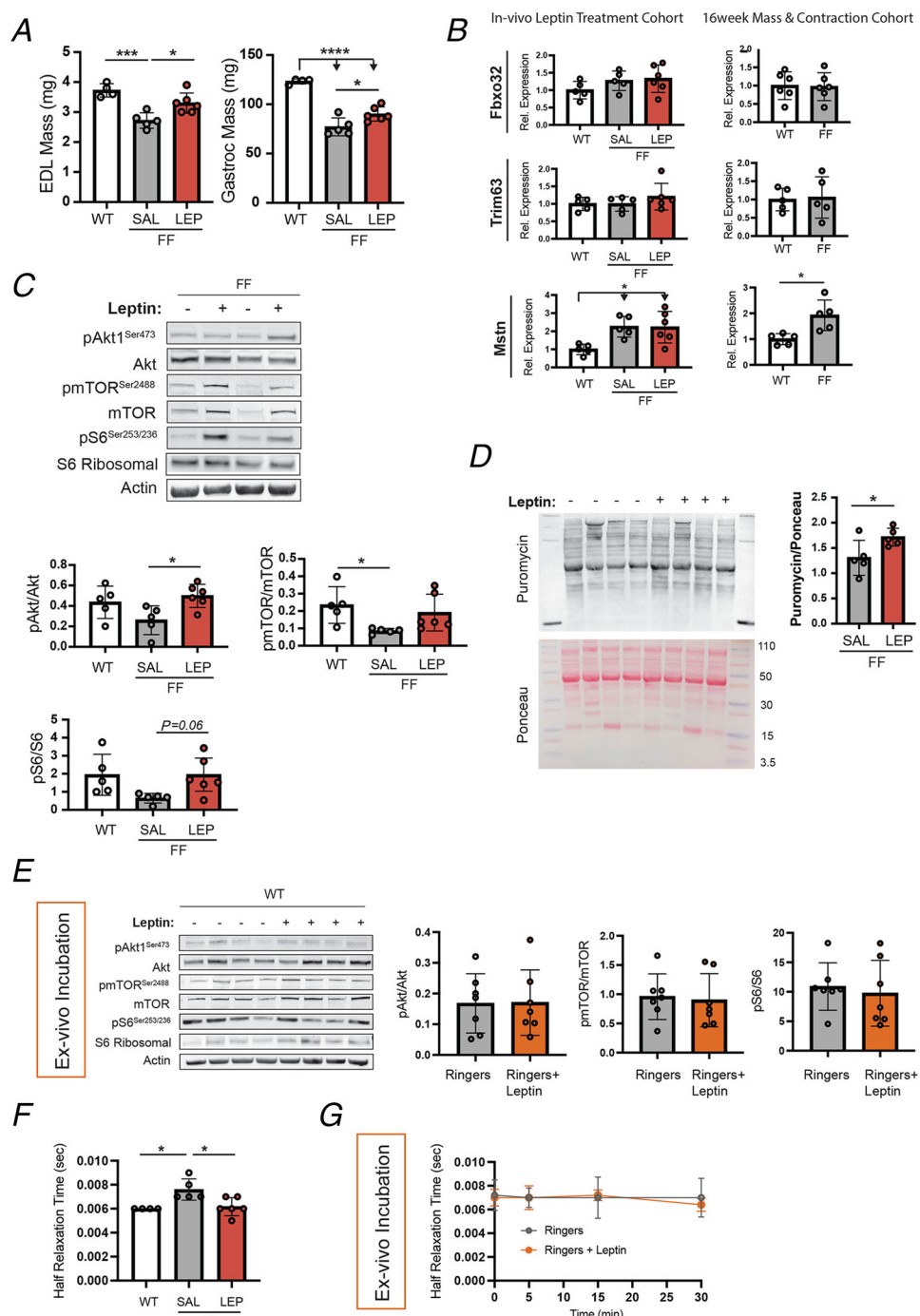

**Figure 8. Exogenous leptin treatment increases muscle protein synthesis *in vivo* but not *ex vivo***
*A*, seven days of leptin injections partially rescues mass of FF EDL and gastrocnemius muscles (red) compared with saline treated FF (grey) and WT littermate controls (white). *B*, mass increases with leptin injections are not accompanied by changes in MAFbx (*Fbxo32*), MuRF1 (*Trim63*) or myostatin (*Mstn*) gene expression. *Fbxo32* and *Trim63* expression is not different between WT and saline treated FF or different between untreated WT and FF (right panel); however, Mstn is significantly upregulated in both cohorts. *C*, abundance of signalling mediators of hypertrophy (pAkt, pmTOR, pS6 ribosomal protein) are elevated in leptin treated FF mice; however, this only reaches significance for pAkt. *D*, puromycin abundance is higher in leptin treated FF muscle compared with saline treatment. *E*, incubation of EDL muscles with leptin *ex vivo* fails to activate signalling mediators of hypertrophy. *F*, prolonged half-relaxation time in twitch contractions is rescued by *in vivo* leptin treatment. *G*, incubation of EDL muscles with leptin *ex vivo* also fails to rescue the prolonged twitch relaxation. Grouped data compared with one-way ANOVA with Tukey's *post hoc* test, *$P < 0.05$, **$P < 0.01$, ***$P < 0.005$, ****$P < 0.001$.

Simha et al., 2021). Our MEF treatment data suggest an explanation for this discrepancy. Injected MEFs formed fat pads of varying size from ∼2% to 20% of the typical total adipose mass of an adult mouse (Collins et al., 2020; Sackmann-Sala et al., 2012), suggesting that only a small amount of fat is required to maintain muscle mass. Interestingly, muscle mass did not correlate with fat pad mass. This suggests that the maintenance of muscle mass by adipose tissue is not dose dependent even below 10% of normal adipose mass and thus an exceedingly small quantity of adipose tissue, similar to residual adipose in the most extensive lipodystrophies (Akinci et al., 2017; Cortes et al., 2009), may be capable of preventing the pathology.

Using the approach of generating transgenic MEFs from leptin knockout mice, we demonstrated that the rescue of muscle mass by WT MEF in FF mice is mediated by leptin. Although traditional approaches to studying the effects of leptin on muscle metabolism have used recombinant leptin delivered by injection or pump, we chose MEF treatment to specifically assess the role of adipose-secreted leptin. Serum leptin concentrations from MEF-rescued mice ranged from ∼5% to 50% of WT values and, similar to fat pad mass, are not correlated with EDL muscle mass. This suggests that the regulation of muscle mass by leptin is not dose dependent and as little as 5% of circulating leptin is capable of restoring muscle mass and contractile performance. This finding offers an explanation for the conflicting data on the effect of leptin administration on muscle mass in *ob/ob* mice and hypoleptinaemic and lipodystrophic humans. Unlike the lipodystrophic models, the muscle phenotype of *ob/ob* mice mirrors the FF model presented here – fast-fibre specific atrophy and slowing of contractile dynamics (Bruton et al., 2002) – presumably due to a complete lack of functional leptin. Fast-fibre specific atrophy is atypical of disuse models (e.g. immobilization, denervation), but frequently found in conjunction with metabolic disturbance (e.g. diabetes, ageing) where leptin resistance has been implicated (Wang & Pessin, 2013). The mechanism for this specificity remains poorly defined, but likely derives from fibre type specificity in the downstream pathways targeted by different stimuli. Delivery of leptin to *ob/ob* mice increases muscle mass and fibre cross sectional area (Mao et al., 2013; Sainz et al., 2009). However, similar delivery of leptin to hypoleptinaemic individuals (Brinkoetter et al., 2011) and lipodystrophic individuals (Moran et al., 2004) with measurable quantities of circulating leptin does not affect lean mass, despite having dramatic effects on adipose mass in the former group and insulin resistance in the latter. It is possible that these individuals with partial adipose tissue can generate sufficient adipose-derived leptin to maintain its action on muscle mass and thus further increases have no further effect.

The mechanism of action of leptin on muscle mass is not well understood and conflicting. Some evidence supports a direct anabolic action of leptin on muscle. Treatment of C2C12 derived myotubes with leptin increases protein synthesis and decreases degradation, which mirrors leptin treatment in *ob/ob* muscle *in vivo* (Mao et al., 2013). Leptin has also been shown to increase myoblast proliferation in primary cultures (Arounleut et al., 2013) and this effect is thought to underlie the increase in sarcopenic muscle mass with leptin treatment (Hamrick et al., 2010). However, leptin is a pleiotropic cytokine with documented effects on nearly every tissue in the body – many of which also play a role in maintaining muscle physiology – suggesting a likely indirect action in addition to direct action. Such a mechanism has been demonstrated for leptin's action on skeletal muscle fatty acid oxidation, where early activation of AMP-activated protein kinase is mediated directly by leptin and later activation is indirect through sympathetic stimulation (Minokoshi et al., 2002). While we have not defined a specific mechanism by which leptin maintains muscle mass here, we have taken advantage of the fact that full rescue of the FF muscle phenotype was accomplished without full rescue of metabolic disturbance in FF mice.

Insulin plays a central role in the maintenance of muscle mass by promoting protein synthesis and inhibiting autophagy and proteolysis. Treatment of *ob/ob* mice with physiological doses of leptin (to match WT serum levels) reverses insulin resistance (Pelleymounter et al., 1995), as it does in lipodystrophic mice and humans (Petersen et al., 2002; Shimomura et al., 1999), suggesting this could be the mechanism of action of muscle mass accretion with leptin treatment. Since WT MEF transplant was able to fully rescue muscle mass without normalizing insulin sensitivity, the rescue mechanism was likely not indirect through this pathway. Furthermore, insulin sensitivity was similar between MEF and MEF-OB groups, while muscle mass and function were significantly different. Other metabolic factors have been reported to be restored by the administration of systemic leptin. For example, physiological doses of leptin also reduce circulating glucocorticoids in *ob/ob* mice (Liu et al., 2016; Pralong et al., 1998) and independent reduction of circulating glucocorticoids in *ob/ob* mice by adrenalectomy increases muscle mass (Saito & Bray, 1984), consistent with the known catabolic action of glucocorticoids on muscle (Shimizu et al., 2011). However, despite elevated plasma levels of the glucocorticoid corticosterone in FF mice, adrenalectomy failed to rescue muscle mass indicating that the MEF transplant rescue was not mediated by a reduction in glucocorticoids.

Finally, chronically elevated circulating triglycerides and FFA cause excessive ectopic storage of lipid (steatosis) in the liver and skeletal muscle, which is proposed to cause lipotoxicity and mitochondrial dysfunction in *ob/ob*

mice (Holmstrom et al., 2013), an effect which is globally associated with reduced protein synthesis and muscle mass (Masgrau et al., 2012). By providing a small adipose depot, both MEF and MEF-OB treatment significantly reduced plasma triglycerides and free fatty acids in FF mice to levels comparable to WT. However, while MEF treatment results in the expected subsequent reduction in muscle intramyocellular lipid and liver mass, MEF-OB treatment elicits only a modest decrease. This is likely due to an additional direct effect of leptin on lipid partitioning in muscle and liver (Huang et al., 2006; Muoio et al., 1997). It is unlikely that lipotoxicity of muscle fibres is responsible for the loss of mass and peak contractile tension in the EDL because the soleus contains a larger fraction of the affected 2a and 2x fibres and exhibits a milder phenotype, but it cannot be completely ruled out as these muscles perform different functions which involve different metabolic profiles (Leijendekker & Elzinga, 1990). Lipotoxicity in the liver could also impact muscle physiology, and future work aims to explore this possibility in further detail. Recent studies have shown a significant impact of adipose ablation on bone growth using this mouse model (Zhang et al., 2021; Zou et al., 2019), illustrating a pathway of adipose–bone crosstalk. Similar to the work of Zhang et al. we find an ∼5% decrease in tibia length in FF mice which is rescued by MEF treatment, but not by MEF-OB. This length discrepancy is smaller than the mass differences in fast fibred muscles (20–30%) and would be expected to equally affect parallel fibred muscles that span the length of the tibia (e.g. plantaris and soleus), whose mass is differentially affected in FF mice. Thus, it is unlikely that growth restriction by decreased skeletal size is fully responsible for the FF muscle mass phenotype, but it is important to note that another musculoskeletal tissue is responsive to MEF treatment and fails to respond to MEF-OB treatment. An adipose–liver–muscle axis or adipose–bone–muscle axis may still be the underlying mechanism, which warrants further investigation.

In addition to narrowing relevant downstream mechanisms, this study has also shed light on some upstream factors. First, though the *ob/ob* mouse has been pivotal in separating the role of increased adipose *vs.* whole body leptin in the physiology of many target tissues, it is unable to triangulate the specific importance of various sources of leptin because of the complete absence of all leptin. While adipose is typically considered to be the major source of circulating leptin, some data support expression and secretion of leptin by muscle (Wang et al., 1998; Wolsk et al., 2012) where it could play an autocrine role. The lack of detectable plasma leptin in FF mice indicates that adipose is the sole source of circulating leptin and that local leptin secretion by muscle is unable to maintain normal muscle mass. In fact, because the muscle pathology in FF mice so closely

mirrors that in the *ob/ob* mouse, muscle-secreted leptin is unlikely to play any major role in muscle physiology. Furthermore, we were able to fully rescue the muscle phenotype with reconstitution of an anatomically distant adipose depot suggesting that paracrine signalling from local adipose tissue (e.g. intramuscular adipose) is not required to maintain muscle mass and strength. Second, the consistent muscle phenotype between *ob/ob* knockout mice and FF mice supports the notion that adipose-derived leptin is the predominant adipokine regulating muscle mass and contractile dynamics. While the FF mouse lacks all adipose secretions, the *ob/ob* mouse has an abundance of adipose and thus ample levels of most adipokines including visfatin and resistin (Favero et al., 2015) suggesting that these cannot compensate for loss of leptin. Two notable exceptions are adiponectin and adipsin. Adiponectin and adipsin secretion decrease with obesity and the *ob/ob* mouse has drastically reduced circulating levels compared with lean mice (Favero et al., 2015). While we find a mild reduction in secreted adiponectin from MEF-OB compared with MEF, it does not impact circulating adiponectin. We did not compare circulating adipsin levels between MEF and MEF-OB treated mice, but examination of mice that constitutively lack adipsin (factor D knockout) did not reveal a muscle phenotype, suggesting reduced adipsin does not play a role in FF muscle pathology. Of course, not all adipokines are beneficial to muscle physiology, and *ob/ob* mice have high adipose expression of a number of pro-inflammatory adipokines known to reduce muscle mass or insulin signalling including IL-6, TNF-$\alpha$, MCP-1 and IL-1$\alpha$/$\beta$ (Fong et al., 1989; Haddad et al., 2005; Sell, Dietze-Schroeder, Kaiser, Eckel et al., 2006; Ye et al., 2007). However, our CM screen did not find elevated secretion of these factors from MEF-OB compared with MEF, suggesting they are not likely driving an independent pathology in MEF-OB treated mice. These intrinsic differences between *ob/ob* adipose and MEF-OB highlight the benefit of our tissue engineering approach as we are able to avoid these pathological adaptations that occur in the native *ob/ob* adipose.

The literature supports several potential mechanisms by which leptin could directly or indirectly regulate muscle mass. Evidence in WT and *ob/ob* mice shows that leptin treatment activates Akt signalling in skeletal muscle (Maroni et al., 2003; Roman et al., 2010). Downstream, activated Akt can affect muscle mass either through inhibiting degradation, predominantly through the ubiquitin ligases MAFbx/atrogin-1 and MuRF1, or through promoting synthesis, predominantly through activation of mTOR. While evidence in *ob/ob* mice suggests that leptin treatment decreases MAFbx/atrogin-1 and MuRF1 expression (Sainz et al., 2009), we did not find this to be the case with treatment of FF mice. The difference could be due to the mouse model (*ob/ob*

mice have unique metabolic dysfunction) or treatment (those mice were treated with more leptin for longer). We cannot rule out these atrogenes as a target of leptin under different conditions as we did not assess dose–response or various time points, but this does suggest that hypercortisolism-mediated upregulation of atrogenes is not responsible for decreased muscle mass in FF mice. Under our conditions, we find increased protein synthesis with leptin treatment suggesting that protein synthesis is, at least in part, driving mass gains. This is supported by experiments in myotubes and *ob/ob* mice (Mao et al., 2013). However, increases in protein synthesis in leptin treated FF mice were relatively moderate and increases in phospho-mTOR and phospho-S6 ribosomal protein did not reach statistical significance suggesting another pathway such as autophagy could also be involved.

There is little data to guide a hypothesis as to how leptin impacts contraction. On the whole, few studies have reported rigorous muscle contractile characterization in a leptin-deficient model. Two studies have reported detailed functional testing of *ob/ob* EDL muscles and did not find a significant decrease in peak tetanic tension (Bruton et al., 2002; Warmington et al., 2000). However, the absolute value of the specific tension was lower in both studies than reported here (∼150–200 kPa *vs.* ∼300 kPa) and lower than previously reported for WT mouse EDL (Luff, 1981), which suggests that muscles may not have reached full activation in these studies. However, both studies are in agreement with this report that absence of leptin induces a significant slowing of relaxation following twitch (increased half-relaxation time). Detailed measurements of calcium transients in fibres isolated from *ob/ob* EDL muscles suggest that there is no deficit in calcium pumping in the sarcoplasmic reticulum, and the authors instead suggest that the kinetics of cross-bridge detachment are altered (Bruton et al., 2002). However, a similar study in the leptin receptor knockout mouse (*db/db*) finds a decrease in peak calcium levels associated with a decrease in peak tetanic tension and prolongation of twitch relaxation in the EDL (Eshima et al., 2017). In this study, we find that *ex vivo* incubation of EDL muscles with leptin fails to rescue the prolongation of twitch relaxation which suggests the mechanism is not direct, but the upstream effectors remain unknown.

In summary, leptin is one of the most pleiotropic cytokines in the body. It has a well-characterized indirect and direct role in regulating muscle metabolism and, in high doses, is implicated in the progression of type 2 diabetes. But outside of obesity, diabetes and the sequellae of metabolic disturbance, little is known about the impact of leptin on muscle homeostasis. Most of our assumptions are inferred from the leptin knockout mouse (*ob/ob*) – which is obese and diabetic. Here we show that not only is leptin required to achieve normal mass and contraction in skeletal muscle, but that requirement is independent, or at least separable, from leptin's role in normalizing systemic metabolism. We further show that despite the fact that this regulation requires less than 10% of normal circulating leptin, no other source of leptin in the body is sufficient to compensate for the lack of adipose-derived leptin. Together these data support a requirement for adipose-derived leptin in the regulation of muscle mass and contraction.

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

## Additional information

### Data availability statement

All data generated or analysed during this study are included in this published article [and its supplementary information files].

### Competing interests

The authors have no competing interests or conflicts of interest related to this work.

### Author contributions

G.A.M., K.H.C., C.A.H. and F.G. were responsible for the conception and design of the study. G.A.M., K.H.C., C.G., E.V.E. and K.L.L. contributed to acquisition, analysis and interpretation of the data. G.A.M. and K.H.C. were responsible for the drafting of the manuscript and G.A.M., C.G., E.V.E., K.L.L., C.A.H. and F.G. all contributed to critical revision of the manuscript. All authors have read and approved the final version of this manuscript and agree to be accountable for all aspects of the work in ensuring that questions related to the accuracy or integrity of any part of the work are appropriately investigated and resolved. All persons designated as authors qualify for authorship, and all those who qualify for authorship are listed.

### Funding

K.H.C., E.V.E. and F.G. are funded through Shriners Hospitals for Children, Musculoskeletal Research Centre Pilot Grant P30 AR074992, and Feasibility and Just-In-Time Grant, Training Grants T32 DK108742 and T32 DK007120. This study was supported, in part, by NIH Grants, AR075773, AG15768, AG46927, AR072999, AR073752.

### Acknowledgements

The authors thank Dr John Atkinson and Dr Xiaobo Wu for the adipsin/complement factor D constitutive knockout mice and Simon Schenk for his insight and guidance.

## Author's present address

C. A. Harris: Early Clinical Development & Experimental Sciences, Regeneron, Pharmaceuticals, Tarrytown, NY, USA.

## Keywords

adipokines, anabolic, fat-muscle cross-talk, muscle contraction

## Supporting information

Additional supporting information can be found online in the Supporting Information section at the end of the HTML view of the article. Supporting information files available:

**Peer Review History**
**Statistical Summary Document**

