## [Peer Review History · The Journal of Physiology]

Leptin Mediates the Regulation of Muscle Mass and Strength by Adipose Tissue

Kelsey H Collins, Chang Gui, Erica V Ely, Kristin L Lenz, Charles A Harris, Farshid Guilak, and Gretchen A Meyer
DOI: 10.1113/JP283034

Corresponding author(s): Gretchen Meyer (meyerg@wustl.edu)

The following individual(s) involved in review of this submission have agreed to reveal their identity: Sue C Bodine (Referee #1)

Review Timeline:

Submission Date:	25-Feb-2022
Editorial Decision:	23-Mar-2022
Revision Received:	27-May-2022
Accepted:	04-Jul-2022

Senior Editor: Scott Powers

Reviewing Editor: Troy Hornberger

Transaction Report:

Dear Dr Meyer,

Re: JP-RP-2022-283034 "Leptin Mediates the Regulation of Muscle Mass and Strength by Adipose Tissue" by Kelsey H Collins, Chang Gui, Erica V Ely, Kristin L Lenz, Charles A Harris, Farshid Guilak, and Gretchen A Meyer

Thank you for submitting your manuscript to The Journal of Physiology. It has been assessed by a Reviewing Editor and by 2 expert Referees and I am pleased to tell you that it is considered to be acceptable for publication following satisfactory revision.

The reports are copied at the end of this email. Please address all of the points and incorporate all requested revisions, or explain in your Response to Referees why a change has not been made.

NEW POLICY: In order to improve the transparency of its peer review process The Journal of Physiology publishes online as supporting information the peer review history of all articles accepted for publication. Readers will have access to decision letters, including all Editors' comments and referee reports, for each version of the manuscript and any author responses to peer review comments. Referees can decide whether or not they wish to be named on the peer review history document.

Authors are asked to use The Journal's premium BioRender (<https://biorender.com/>) account to create/redraw their Abstract Figures. Information on how to access The Journal's premium BioRender account is here: <https://physoc.onlinelibrary.wiley.com/journal/14697793/biorender-access> and authors are expected to use this service. This will enable Authors to download high-resolution versions of their figures. The link provided should only be used for the purposes of this submission. Authors will be charged for figures created on this premium BioRender account if they are not related to this manuscript submission.

I hope you will find the comments helpful and have no difficulty returning your revisions within 4 weeks.

Your revised manuscript should be submitted online using the links in Author Tasks Link Not Available.

Any image files uploaded with the previous version are retained on the system. Please ensure you replace or remove all files that have been revised.

REVISION CHECKLIST:

- Article file, including any tables and figure legends, must be in an editable format (eg Word)
- Abstract figure file (see above)
- Statistical Summary Document
- Upload each figure as a separate high quality file
- Upload a full Response to Referees, including a response to any Senior and Reviewing Editor Comments;
- Upload a copy of the manuscript with the changes highlighted.

- A potential 'Cover Art' file for consideration as the Issue's cover image;
- Appropriate Supporting Information (Video, audio or data set https://jp.msubmit.net/cgi-bin/main.plex?form_type=display_requirements#supp).

To create your 'Response to Referees' copy all the reports, including any comments from the Senior and Reviewing Editors, into a Word, or similar, file and respond to each point in colour or CAPITALS and upload this when you submit your revision.

I look forward to receiving your revised submission.

If you have any queries please reply to this email and staff will be happy to assist.

Yours sincerely,

Scott K. Powers
Senior Editor
The Journal of Physiology
<https://jp.msubmit.net>
<http://jp.physoc.org>
The Physiological Society
Hodgkin Huxley House
30 Farringdon Lane
London, EC1R 3AW
UK
<http://www.physoc.org>
<http://journals.physoc.org>

REQUIRED ITEMS:

-Author photo and profile. First (or joint first) authors are asked to provide a short biography (no more than 100 words for one author or 150 words in total for joint first authors) and a portrait photograph. These should be uploaded and clearly labelled with the revised version of the manuscript. See Information for Authors for further details.

-You must start the Methods section with a paragraph headed Ethical Approval. A detailed explanation of journal policy and regulations on animal experimentation is given in Principles and standards for reporting animal experiments in The Journal of Physiology and Experimental Physiology by David Grundy J Physiol, 593: 2547-2549. doi:10.1113/JP270818.). A checklist outlining these requirements and detailing the information that must be provided in the paper can be found at: <https://physoc.onlinelibrary.wiley.com/hub/animal-experiments>. Authors should confirm in their Methods section that their experiments were carried out according to the guidelines laid down by their institution's animal welfare committee, and conform to the principles and regulations as described in the Editorial by Grundy (2015). The Methods section must contain details of the anaesthetic regime: anaesthetic used, dose and route of administration and method of killing the experimental animals.

-Your manuscript must include a complete Additional Information section

-Please upload separate high-quality figure files via the submission form.

-A Statistical Summary Document, summarising the statistics presented in the manuscript, is required upon revision. It must be on the Journal's template, which can be downloaded from the link in the Statistical Summary Document section here: https://jp.msubmit.net/cgi-bin/main.plex?form_type=display_requirements#statistics

-Papers must comply with the Statistics Policy https://jp.msubmit.net/cgi-bin/main.plex?form_type=display_requirements#statistics

In summary:

-If $n \leq 30$, all data points must be plotted in the figure in a way that reveals their range and distribution. A bar graph with data points overlaid, a box and whisker plot or a violin plot (preferably with data points included) are acceptable formats.

-If $n > 30$, then the entire raw dataset must be made available either as supporting information, or hosted on a not-for-profit repository e.g. FigShare, with access details provided in the manuscript.

- n clearly defined (e.g. x cells from y slices in z animals) in the Methods. Authors should be mindful of pseudoreplication.

-All relevant n values must be clearly stated in the main text, figures and tables, and the Statistical Summary Document (required upon revision)

-The most appropriate summary statistic (e.g. mean or median and standard deviation) must be used. Standard Error of the Mean (SEM) alone is not permitted.

-Exact p values must be stated. Authors must not use 'greater than' or 'less than'. Exact p values must be stated to three significant figures even when 'no statistical significance' is claimed.

-Statistics Summary Document completed appropriately upon revision

-Please include an Abstract Figure. The Abstract Figure is a piece of artwork designed to give readers an immediate understanding of the research and should summarise the main conclusions. If possible, the image should be easily 'readable' from left to right or top to bottom. It should show the physiological relevance of the manuscript so readers can assess the importance and content of its findings. Abstract Figures should not merely recapitulate other figures in the manuscript. Please try to keep the diagram as simple as possible and without superfluous information that may distract from the main conclusion(s). Abstract Figures must be provided by authors no later than the revised manuscript stage and should be uploaded as a separate file during online submission labelled as File Type 'Abstract Figure'. Please ensure that you include the figure legend in the main article file. All Abstract Figures should be created using BioRender. Authors should use The Journal's premium BioRender account to export high-resolution images. Details on how to use and access the premium account are included as part of this email.

EDITOR COMMENTS

Reviewing Editor:

Your manuscript has now been reviewed by two referees with expertise in adipose and muscle physiology. Both referees considered the work to be well designed, executed, and written. Overall there was a good deal of interest in the study, but a major concern was raised. Specifically, it was noted that the study does not offer insight into how Leptin exerts its effects on muscle mass and strength. I agree with this concern and believe that the impact of your work would be substantially improved if you can provide some basic insight into Leptin's mechanism of action (e.g., changes in proteins synthesis, protein degradation, etc.). Unfortunately, without this kind of insight, the potential impact of your study would not be rated high enough to warrant publication.

Senior Editor:

Thank you for submitting your work to the Journal of Physiology. Your report has undergone a careful evaluation by two expert referees and a review editor (RE). The consensus opinion is that your study is interesting and provides new information. However, a shortcoming defined by one of the referees and the RE is the lack of data that directly connects leptin to increased muscle mass. Nonetheless, I invite the authors to revise their report in accordance with reviewer suggestions. We look forward to receiving a revised manuscript.

REFEREE COMMENTS

Referee #1:

The manuscript by Collins et al. examines the role of adipokines in regulating skeletal muscle mass. The study used lipodystrophic fat-free mice to show that adipose tissue is necessary for the development of normal muscle mass and strength. The deficits in mass and strength were rescued by adipose-derived leptin. The experiments are well designed and the analysis of muscle size and contractile properties is comprehensive and rigorous. The data are interesting and convincingly demonstrate that adipose-derived leptin is important in the regulation of skeletal muscle mass and strength, specifically of the type IIb fibers. The one limitation of the study is that it does not address the mechanism by which leptin increases muscle mass and strength and why the selectivity which regard to fiber type. Is there an increase in protein synthesis within the type IIb fibers? There is some literature suggesting that leptin suppresses certain atrophy associated genes such as MurF1 and MAFbx/atrogen1.

Minor comments:

Figure 4 legend: there are 2 C's. Was the whole gastrocnemius complex analyzed? How many fibers were analyzed?

For terminal experiments (contractile testing), were the animals anesthetized prior to removal of the muscles? How were the mice euthanized?

Referee #2:

The manuscript by Collins et al addresses the role of adipose tissue in maintaining muscle mass and the molecular mechanisms. They identify leptin as the sole mediator of the adipose-muscle crosstalk that regulates muscle mass. A number of possible mechanisms by which leptin may act are explored. This is a well designed and executed study, and the conclusions are supported by the data.

Comment

Given that there were different results for adiponectin in CM versus serum (Fig 5E versus F), it would be useful, if possible, to assess LIF and IL15 in serum as the authors did for adiponectin.

Minor:

For the statement in results "Glucose tolerance was similar in all groups (Fig. 6A).", could the authors check the p-value for MEF versus MEF-OB. I am surprised this was not significant.

END OF COMMENTS

Confidential Review

25-Feb-2022

Leptin Mediates the Regulation of Muscle Mass and Strength by Adipose Tissue

Kelsey H Collins, Chang Gui, Erica V Ely, Kristin L Lenz, Charles A Harris, Farshid Guilak, and Gretchen A Meyer

Thank you very much for your careful consideration of the manuscript and for your insightful and helpful comments. We appreciate the time that you have put into your evaluation and respond to each of your concerns below on a point-by-point basis. We believe these changes strengthen the manuscript

Reviewing Editor

Your manuscript has now been reviewed by two referees with expertise in adipose and muscle physiology. Both referees considered the work to be well designed, executed, and written. Overall there was a good deal of interest in the study, but a major concern was raised. **Specifically, it was noted that the study does not offer insight into how Leptin exerts its effects on muscle mass and strength. I agree with this concern and believe that the impact of your work would be substantially improved if you can provide some basic insight into Leptin's mechanism of action (e.g., changes in proteins synthesis, protein degradation, etc.).** Unfortunately, without this kind of insight, the potential impact of your study would not be rated high enough to warrant publication.

Thank you for the positive review of our work. We agree that uncovering leptin's mechanism of action on muscle mass and strength will be key to fully realizing its therapeutic potential. While the question of central vs. peripheral action of leptin on muscle will require extensive experimentation beyond the scope of this study, you are correct that we can shed some light on the muscle pathways affected by leptin (whether direct or indirect). To this end, we have treated additional FF experimental mice with leptin for 7 days and then probed protein synthesis and degradation pathways. We have provided the following additional insights in this manuscript:

- 1) Short term administration of exogenous leptin increases muscle mass in FF mice and rescues the prolongation of twitch tension
- 2) The increased muscle mass is not driven by suppression of atrogene expression (MuRF1 and MAFbx/atrogen1), as suggested by Reviewer 1. Expression of MuRF1 (Trim63) and MAFbx (Fbxo32) were not affected by leptin administration and were not different between WT and FF mice.
- 3) FF mice have elevated expression of Myostatin which could contribute to their decreased muscle mass, but this was not affected by leptin administration suggesting that leptin is acting through a different mechanism
- 4) Leptin administration increased phosphorylation of the signaling mediators that drive muscle hypertrophy including mTOR, Akt and S6 ribosomal protein. It also increased protein synthesis as assessed by puromycin labeling. Together this suggests that leptin increases muscle mass at least in part by increasing protein synthesis.
- 5) The increase in mTOR/Akt/S6 phosphorylation and prolongation of twitch tension could not be replicated by ex-vivo incubation of muscle in leptin suggesting that these effects are not driven directly by acute leptin exposure.

We hope that these insights in addition to the mechanistic work identifying leptin as the sole adipokine mediating muscle mass and strength provide sufficient impact for the Journal of Physiology. These data can be found in figure 8 of the manuscript.

Reviewer #1:

The manuscript by Collins et al. examines the role of adipokines in regulating skeletal muscle mass. The study used lipodystrophic fat-free mice to show that adipose tissue is necessary for the development of normal muscle mass and strength. The deficits in mass and strength were rescued by adipose-derived leptin. The experiments are well designed and the analysis of muscle size and contractile properties is comprehensive and rigorous. The data are interesting and convincingly demonstrate that adipose-derived leptin is important in the regulation of skeletal muscle mass and strength, specifically of the type IIb fibers. **The one limitation of the study is that it does not address the mechanism by which leptin increases muscle mass and strength and why the selectivity which regard to fiber type. Is there an increase in protein synthesis within the type IIb fibers? There is some literature suggesting that leptin suppresses certain atrophy associated genes such as MuRF1 and MAFbx/atrogin1.**

Thank you for your thoughtful review. In response to your suggestion, we have added additional experiments aimed at elucidating the mechanism by which leptin increases muscle mass. Specifically, we treated FF mice with leptin and evaluated changes in hypertrophic and atrophic signaling markers. Surprisingly, we did not find any changes in MuRF1 or MAFbx gene expression with leptin treatment. And perhaps more surprisingly, expression of these genes was not different between WT and FF mice in general. We used cachectic samples as a positive control to validate the primers and confirmed the lack of difference between WT and FF in some of our previous samples, so we feel confident in this result. Our data suggest that protein synthesis is a target of leptin. Phosphorylation of mTOR, Akt and S6 Ribosomal protein were all increased with leptin administration. Further, protein synthesis was increased as assayed by the SUNSET technique. While this does not rule out a role for other mediators of protein degradation (e.g. 20s proteasome, autophagy), it indicates that protein synthesis is involved in the mass increase. We also performed ex-vivo leptin incubations of the EDL muscles of saline treated FF mice, but failed to find the same increase in Akt/mTOR/S6 signaling suggesting the mechanism may not be acute direct leptin action. We have incorporated this data into the manuscript in a new Figure 8.

In these experiments, we focused on the muscles that are predominately type IIb fibers (EDL and white gastroc) to assess the response of this fiber type. The selectivity of type IIb fiber atrophy in FF muscle aligns with mouse models of cachexia, sepsis and diabetes. The mechanisms driving the fiber type specificity are hypothesized to derive from differences in the relative sensitivity of the different fiber types to different stimuli, but are poorly defined otherwise. We have added a note to this effect to the Discussion.

1. Figure 4 legend: there are 2 C's. Was the whole gastrocnemius complex analyzed? How many fibers were analyzed?

Thanks for noting this error. The second (C) is now a (D).

Only a portion of the gastrocnemius complex was analyzed. Representative images were taken in defined regions (Soleus mid-belly, Plantaris mid-belly, red gastrocnemius and white gastrocnemius). Each image comprised 100-200 fibers. For the analysis of fiber area by fiber type, fibers of same type were grouped together yielding >50 fibers per type for the histogram and mean fiber area analyses. These details have now been added to the Methods section.

2. For terminal experiments (contractile testing), were the animals anesthetized prior to removal of the muscles? How were the mice euthanized?

Yes, the animals were fully anesthetized with 2% inhaled isoflurane at 2 L/min before any incisions. This is a standard protocol which does not alter muscle physiology. This information and the method of euthanization have now been added to the Methods section.

Reviewer #2:

The manuscript by Collins et al addresses the role of adipose tissue in maintaining muscle mass and the molecular mechanisms. They identify leptin as the sole mediator of the adipose-muscle crosstalk that regulates muscle mass. A number of possible mechanisms by which leptin may act are explored. This is a well designed and executed study, and the conclusions are supported by the data.

Thank you. And thank you for your positive comments and review.

1. Given that there were different results for adiponectin in CM versus serum (Fig 5E versus F), it would be useful, if possible, to assess LIF and IL15 in serum as the authors did for adiponectin.

We agree with this suggestion. Unfortunately, we don't have sufficient serum samples remaining for this. We used most of the serum for a Luminex multiplex, but those results contained an artifact that called those results into question and so they weren't included in this manuscript.

2. For the statement in results "Glucose tolerance was similar in all groups (Fig. 6A).", could the authors check the p-value for MEF versus MEF-OB. I am surprised this was not significant.

Thank you for noting this error. In fact, in the GTT AUC analysis, there is a significant difference between MEF and MEF-OB. The figure and associated text has been updated to reflect this change.

Dear Dr Meyer,

Re: JP-RP-2022-283034R1 "Leptin Mediates the Regulation of Muscle Mass and Strength by Adipose Tissue" by Kelsey H Collins, Chang Gui, Erica V Ely, Kristin L Lenz, Charles A Harris, Farshid Guilak, and Gretchen A Meyer

I am pleased to tell you that your paper has been accepted for publication in The Journal of Physiology.

NEW POLICY: In order to improve the transparency of its peer review process The Journal of Physiology publishes online as supporting information the peer review history of all articles accepted for publication. Readers will have access to decision letters, including all Editors' comments and referee reports, for each version of the manuscript and any author responses to peer review comments. Referees can decide whether or not they wish to be named on the peer review history document.

The last Word version of the paper submitted will be used by the Production Editors to prepare your proof. When this is ready you will receive an email containing a link to Wiley's Online Proofing System. The proof should be checked and corrected as quickly as possible.

Authors should note that it is too late at this point to offer corrections prior to proofing. The accepted version will be published online, ahead of the copy edited and typeset version being made available. Major corrections at proof stage, such as changes to figures, will be referred to the Reviewing Editor for approval before they can be incorporated. Only minor changes, such as to style and consistency, should be made a proof stage. Changes that need to be made after proof stage will usually require a formal correction notice.

All queries at proof stage should be sent to TJP@wiley.com

Are you on Twitter? Once your paper is online, why not share your achievement with your followers. Please tag The Journal (@jphysiol) in any tweets and we will share your accepted paper with our 23,000+ followers!

Yours sincerely,

Scott K. Powers
Senior Editor
The Journal of Physiology
<https://jp.msubmit.net>
<http://jp.physoc.org>
The Physiological Society
Hodgkin Huxley House
30 Farringdon Lane
London, EC1R 3AW
UK
<http://www.physoc.org>
<http://journals.physoc.org>

P.S. - You can help your research get the attention it deserves! Check out Wiley's free Promotion Guide for best-practice recommendations for promoting your work at www.wileyauthors.com/eeo/guide. And learn more about Wiley Editing Services which offers professional video, design, and writing services to create shareable video abstracts, infographics, conference posters, lay summaries, and research news stories for your research at www.wileyauthors.com/eeo/promotion.

*** IMPORTANT NOTICE ABOUT OPEN ACCESS ***

To assist authors whose funding agencies mandate public access to published research findings sooner than 12 months after publication The Journal of Physiology allows authors to pay an open access (OA) fee to have their papers made freely available immediately on publication.

You will receive an email from Wiley with details on how to register or log-in to Wiley Authors Services where you will be able to place an OnlineOpen order.

You can check if your funder or institution has a Wiley Open Access Account here <https://authorservices.wiley.com/author-resources/Journal-Authors/licensing-and-open-access/open-access/author-compliance-tool.html>

Your article will be made Open Access upon publication, or as soon as payment is received.

If you wish to put your paper on an OA website such as PMC or UKPMC or your institutional repository within 12 months of

publication you must pay the open access fee, which covers the cost of publication.

OnlineOpen articles are deposited in PubMed Central (PMC) and PMC mirror sites. Authors of OnlineOpen articles are permitted to post the final, published PDF of their article on a website, institutional repository, or other free public server, immediately on publication.

Note to NIH-funded authors: The Journal of Physiology is published on PMC 12 months after publication, NIH-funded authors DO NOT NEED to pay to publish and DO NOT NEED to post their accepted papers on PMC.

EDITOR COMMENTS

Reviewing Editor:

Well done!!!

Senior Editor:

Thank you for submitting your work to the Journal of Physiology and congratulations on the completion of an excellent study.

REFEREE COMMENTS

Referee #2:

The revised manuscripts addresses my concerns.

1st Confidential Review

27-May-2022